# Fusion algorithm of visible and infrared image based on anisotropic diffusion and image enhancement (capitalize only the first word in a title (or heading), the first word in a subtitle (or subheading), and any proper nouns)

Hui Huang[1], Linlu Dong[2], Zhishuang Xue[1], Xiaofang Liu[1,3]*, Caijian Hua[3]

**1** Artificial Intelligence Key Laboratory of Sichuan Province, Automation and Information Engineering, Sichuan University of Science and Engineering, Zigong, China, **2** School of Information Engineering, Southwest University of Science and Technology, Mianyang, China, **3** School of Computer Science and Engineering, Sichuan University of Science and Engineering, Zigong, China

\* 150507076@qq.com

**Data Availability Statement:** To ensure the reliability of the experimental effect, the infrared

## Abstract

Aiming at the situation that the existing visible and infrared images fusion algorithms only focus on highlighting infrared targets and neglect the performance of image details, and cannot take into account the characteristics of infrared and visible images, this paper proposes an image enhancement fusion algorithm combining Karhunen-Loeve transform and Laplacian pyramid fusion. The detail layer of the source image is obtained by anisotropic diffusion to get more abundant texture information. The infrared images adopt adaptive histogram partition and brightness correction enhancement algorithm to highlight thermal radiation targets. A novel power function enhancement algorithm that simulates illumination is proposed for visible images to improve the contrast of visible images and facilitate human observation. In order to improve the fusion quality of images, the source image and the enhanced images are transformed by Karhunen-Loeve to form new visible and infrared images. Laplacian pyramid fusion is performed on the new visible and infrared images, and superimposed with the detail layer images to obtain the fusion result. Experimental results show that the method in this paper is superior to several representative image fusion algorithms in subjective visual effects on public data sets. In terms of objective evaluation, the fusion result performed well on the 8 evaluation indicators, and its own quality was high.

## 1 Introduction

Due to the limitation of the sensor system, the image information acquired by a single sensor cannot comprehensively describe the target scene. Therefore, it is very important to capture the detailed information of corresponding scenes several times by combining multiple sensors information to capture the target scene. Image fusion is to process the images collected by

image and visible image data selected are all from TNO_Image_Fusion_Dataset (https://figshare.com/articles/dataset/TNO_Image_Fusion_Dataset/1008029) and Li (https://github.com/hli1221/imagefusion_deeplearning/tree/master/IV_images).

**Funding:** This work is supported in part by the Science and Technology Department Project of Sichuan Provincial of China, under Grant 2017GZ0303, in part by Academician (Expert) Workstation Fund Project of Sichuan Province of China, under Grant 2016YSGZZ01, in part by Special Fund for Training High Level Innovative Talents of Sichuan University of Science and Engineering, under Grant B12402005, in part by the Academician Workstation Project of Sichuan Province of China, under Grant 2017YSGZZ04, and Sichuan University of Science and Engineering for Talent introduction project, under Grant 2017RCL59.

**Competing interests:** The authors have declared that no competing interests exist.

multiple source channels in the same scene to maximize the effective information in different channels, improve the utilization of each image, and make the fusion result more comprehensive and clear, which is convenient for people to observe. Image fusion is not only a kind of image enhancement technology, but also an important branch of information fusion. It is a hot spot in information fusion research. It is widely used in remote sensing [1], digital camera [2], military technology [3, 4], objects recognition [5, 6], medical imaging [7] and other fields.

Generally, image fusion is divided into three levels: data-level fusion, feature-level fusion, and decision-level fusion. Data-level fusion is also called pixel-level fusion, which refers to the process of directly processing the data collected by the sensor to obtain a fusion image, make full use of the input data information to better retain the object and background information [8]. Although pixel-level image fusion is the lowest level of fusion, it directly operates on pixels and serves as the basis for other fusion levels, enriches useful information, and the obtained images are consistent with the human visual system [9]. Infrared images are not affected by lighting and camouflage, but the resolution is low. The visible image has high spatial resolution, but it is susceptible to external interference. These two kinds of images have complementary characteristics. The fusion of them is one of the important applications of pixel-level multi-source image fusion [10, 11].

The pixel-level image fusion algorithm includes two kinds of algorithms in the spatial domain and the transform domain. The spatial domain algorithm includes many fusion rules such as gray-scale weighted average method [12], contrast modulation method [13], and Principal Component Analysis(PCA) [7], etc. Transform domain algorithms include Wavelet Transform(WT) method [14, 15], pyramid decomposition fusion algorithm [16], Curvelet Transform(CVT) [17], Non-Subsampled Contourslet Transform(NSCT) [18], Non-Subsampled Shearlet Transform(NSST) [19], etc. The algorithm based on transform domain is the current mainstream fusion algorithm of infrared image and visible image [8]. The main idea is to map the source image from the spatial domain to a sparse transform domain, and corresponding fusion is carried out according to the rules of the transform domain, and the result of image fusion is obtained after inverse transformation. In recent years, many scholars have conducted in-depth research in this area. Chao [20] proposed a wavelet-based image fusion algorithm, which uses high frequency band and low frequency band to perform fusion. Although the fusion efficiency is very high, the fusion results in some direction information are missed. On this basis, Do [21] proposed a wavelet-based contourslet conversion algorithm, which flexibly realizes multi-resolution and directional extensibility, can better preserve directional information. Adu [22, 23] proposed a coefficient selection method based on NSCT, visual and gradient features, which achieves remarkable results in target extraction, and also achieves good results in detail preservation. Ma [24] proposed a gradient transfer fusion algorithm based on gradient transfer and total variation minimization, which can effectively saves the main intensity distribution in the infrared image and the gradient change in the visible image, and it also promotes the ability to correct without pre-registration. Bavirisetti [25] proposed a fusion algorithm that combines Anisotropic Diffusion(AD) and Karhunen-Loeve (K-L) transform. This algorithm can extract detailed layer information according to AD and retain the detailed texture. Fu [26] proposed a fusion algorithm using Robust Principal Component Analysis(RPCA) and NSCT to decompose the image through RPCA to obtain the corresponding sparse matrix, and used the sparse matrix for the fusion of high and low frequency coefficients. This algorithm can effectively highlight infrared targets and retain the background information in the visible image. Huang [27] proposed an NSST algorithm based on different restricted conditions, which fuses high frequency bands through gradient constraints to ensure that the image can get more details. The low frequency bands are merged through saliency constraints to make the target more prominent. Ma [28, 29] first proposed an end-to-end

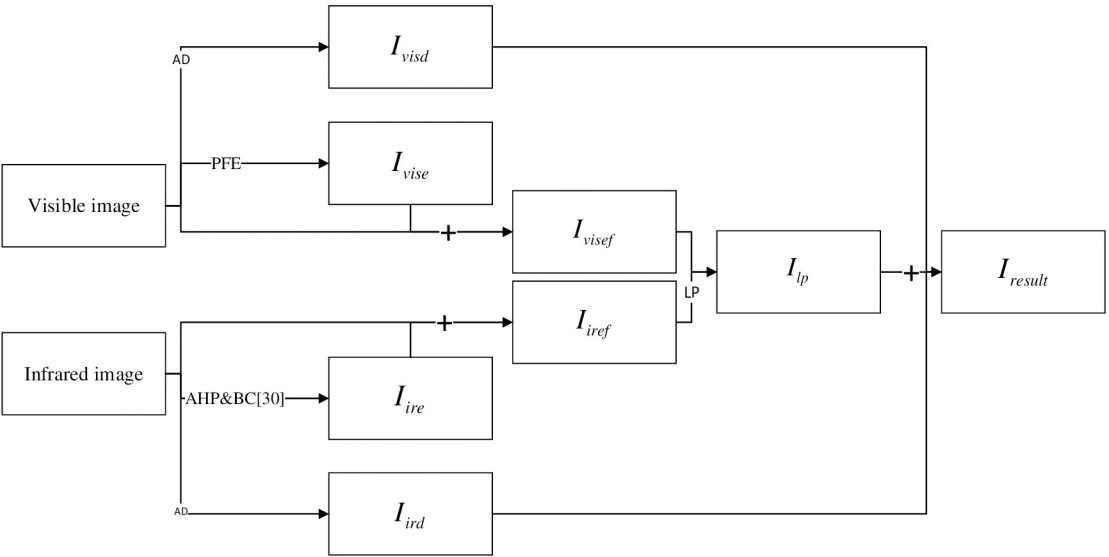

**Fig 1. The structure of this paper.**

infrared and visible image fusion algorithm, which generates a fusion image with main infrared intensity and additional visible gradient under the framework of Generative Adversarial Networks(GAN), and then introduced the detail loss and target edge intensity loss to further enrich the detail texture. Liu [30, 31] first proposed a Convolutional Neural Network (CNN) based deep learning model, which is effectively saves the details of the image, and then a novel fusion scheme based on Non-Subsampled Shearlet Transform (NSST), visual saliency and multi-objective artificial bee colony optimizing spiking cortical mode is proposed. It can be seen that in the fusion research of infrared and visible images, the prominence of infrared targets and the details of visible gradients are of great concern. People hope to improve the resolution and detail richness through various fusion algorithms.

This research focuses on pixel-level image fusion. In order to better integrate the characteristics of infrared images and visible images, this algorithm performs different types of enhancement and fusion on the two source image to highlight their respective advantages. The structure of this paper is shown in Fig 1 The main contributions of the proposed fusion algorithm are as follows:

The main contributions of the proposed fusion algorithm are as follows:

1. The detail layers $I_{ird}$ and $I_{visd}$ of infrared and visible images can be obtain by AD algorithm, to better preserve clear outline and detail information.

2. We propose a Power Function Enhancement (PFE) algorithm to simulate illumination for visible images to obtain visible enhanced images $I_{vise}$, and use Adaptive Histogram Partition (AHP) and Brightness Correction (BC) algorithms to enhance infrared images to obtain infrared enhanced images $I_{ire}$.

3. Based on the K-L transform to fuse the enhanced image and the source image, the $I_{iref}$ results and $I_{visf}$ are used as new visible and infrared source image to fully retain the structural characteristics of the image and improve the fusion quality.

4. In order to better preserve the characteristics of the image and improve the brightness of the image, Laplacian fusion is performed on $I_{iref}$ and $I_{visf}$ to obtain the fused image $I_{lp}$, and the weighted fusion of $I_{lp}$, $I_{ird}$ and $I_{ird}$ is used to obtain the result $I_{result}$.

The main structure of this paper is as follows: section 2 briefly introduces some basic concepts, section 3 proposes a specific image fusion scheme, section 4 provides experimental results and data analysis, and section 5 summarizes the full work.

## 2 Anisotropic diffusion

Anisotropic Diffusion(AD) is also known as the Perona-Malik(P-M) formula, the noise in image is eliminated by Partial Differential Equation(PDE). The image is regarded as a heat field, and each pixel is used as a heat flow. According to the relationship between the current pixel and the surrounding pixels, it is determined whether to diffuse to the surroundings. For example, when the distance between current pixel and surrounding pixel is large, the surrounding pixels may form boundary. Then the current pixel will not diffuse to the boundary, and this boundary is preserved. It overcomes the shortcomings of isotropic diffusion, isotropic diffusion may smooth the background heavily, thus the edge information is lost. The AD equation [32] is proposed:

$$I_t = div(c(x, y, t)\nabla I) = c(x, y, t)\Delta I + \nabla c \cdot \nabla I \tag{1}$$

Where $I_t$ is the source image, $div$ represents the divergence operator, $c(x, y, t)$ represents the flux function or diffusion rate, $\nabla$ represents the gradient operator, $\Delta$ represents the Laplacian operator, $t$ represents the time or the number of iterations, and $\cdot$ represents the value range.

Convert Eq (1) into a thermal equation, use Forward-Time and Central-Space (FTCS) to solve this equation [24], the solution of the partial differential equation is as follows:

$$I_{x,y}^{t+1} = I_{x,y}^t + \lambda(c_N \cdot \bar{\nabla}_N I_{x,y}^t + c_S \cdot \bar{\nabla}_S I_{x,y}^t + c_E \cdot \bar{\nabla}_E I_{x,y}^t + c_W \cdot \bar{\nabla}_W I_{x,y}^t) \tag{2}$$

From Eq (2), we can see that the image $I_{x,y}^{t+1}$ at the higher scale $t+1$ depends on the image $I_{x,y}^t$ at the previous scale $t$, $\lambda$ is a stable constant, and satisfies $0 \leqslant \lambda \leqslant \frac{1}{4}$. $\bar{\nabla}_E$, $\bar{\nabla}_S$, $\bar{\nabla}_W$ and $\bar{\nabla}_N$ represents the difference between the nearest neighbours in the east, south, west, and north directions, respectively, defined as follows:

$$
\begin{aligned}
\bar{\nabla}_E I_{x,y}^t &= I_{x,y+1} - I_{x,y} \\
\bar{\nabla}_S I_{x,y}^t &= I_{x+1,y} - I_{x,y} \\
\bar{\nabla}_W I_{x,y}^t &= I_{x,y-1} - I_{x,y} \\
\bar{\nabla}_N I_{x,y}^t &= I_{x-1,y} - I_{x,y}
\end{aligned}
\tag{3}
$$

Similarly, $c_E$, $c_S$, $c_W$ and $c_N$ represent the flux function or diffusion rate in the east, south, west, and north directions, respectively, which are defined as follows:

$$
\begin{aligned}
c_{E_{x,y}}^t &= g\left(\left\|(\nabla I)_{x,y+{}^1\!/_2}^t\right\|\right) = g(|\bar{\nabla}_E I_{x,y}^t|) \\
c_{S_{x,y}}^t &= g\left(\left\|(\nabla I)_{x-{}^1\!/_2,y}^t\right\|\right) = g(|\bar{\nabla}_S I_{x,y}^t|) \\
c_{W_{x,y}}^t &= g\left(\left\|(\nabla I)_{x,y-{}^1\!/_2}^t\right\|\right) = g(|\bar{\nabla}_W I_{x,y}^t|) \\
c_{N_{x,y}}^t &= g\left(\left\|(\nabla I)_{x+{}^1\!/_2,y}^t\right\|\right) = g(|\bar{\nabla}_N I_{x,y}^t|)
\end{aligned}
\tag{4}
$$

In formula Eq (4), $g(\cdot)$ represents a monotonic decreasing function, where $g(0) = 1$. $g(\cdot)$ can be expressed by different functions, Perona sit [30] proposed two forms as follows:

$$
g(\nabla I) = e^{-\left(\frac{\|\nabla I\|}{K}\right)^2}
\tag{5}
$$

$$
g(\nabla I) = \frac{1}{1+\left(\frac{\|\nabla I\|}{K}\right)^2}
\tag{6}
$$

Where $K$ represents the thermal conductivity or gradient threshold, which is used to control the sensitivity of the edge. The functions in Eqs (5) and (6) provide a trade-off between smoothing and edge preservation. Eq (5) is suitable for images with more high-contrast edges. Eq (6) is suitable for images with large areas covering small areas. For a given image $I$, the AD is denoted as $AD(I)$.

Infrared image and visible source image are represented by $I_{ir}(x, y)$ and $I_{vis}(x, y)$. The base layer images obtained by the infrared image and the visible source image through the aniso-tropic diffusion algorithm are expressed as $I_{ir}^A(x, y)$ and $I_{vis}^A(x, y)$, where $I_{ir}^A(x, y) = AD(I_{ir})$ and $I_{vis}^A(x, y) = AD(I_{vis})$. According to the difference between the source image and the base layer image, the detailed layer images of the infrared image $I_{ird}(x, y)$ and the visible image $I_{visd}(x, y)$ are defined as follows:

$$
\begin{aligned}
I_{ird}(x, y) &= I_{ir}(x, y) - I_{ir}^A(x, y) \\
I_{visd}(x, y) &= I_{vis}(x, y) - I_{vis}^A(x, y)
\end{aligned}
\tag{7}
$$

## 3 The proposed fusion method

### 3.1 Power function enhancement method of visible image

High visibility image details can more clearly reflect the targets in the scene, the image in the process of shooting is influenced by factors such as light and noise, the image visual quality is not satisfactory, then combined with the principle of fitting method, this study aims at dark scene under visible image, this paper presents a simple and practical power function enhance-ment algorithm for visible images in dark scenes. In the fitting problem, it is not necessary for the curve to pass through all given points. The aim of the fitting problem is to find a function curve that is the closest to all data points under a certain criterion, that is, the curve fitting is the best.

Power function is one of the basic elementary functions, which can be used to enhance the image. The formula of PFE is defined as follows:

$$
I_{vise} = C_1 * I_{vis}^{C_2}
\tag{8}
$$

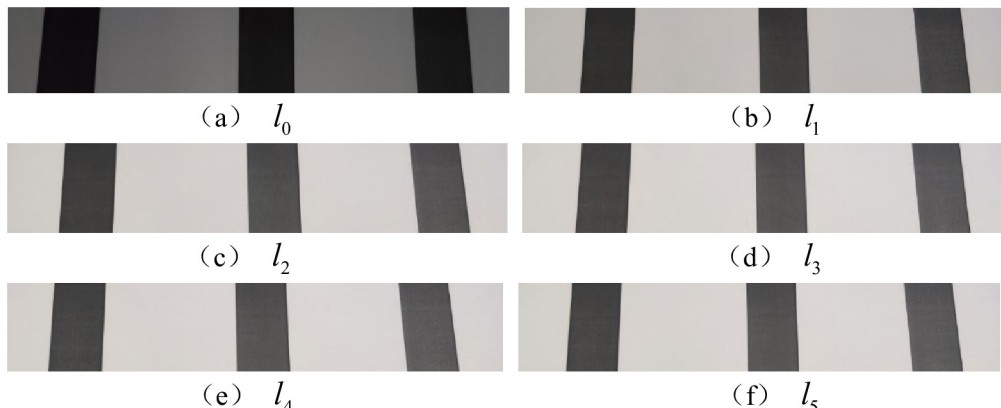

**Fig 2. Results acquired at different brightness stages.** It can be found in Fig 2 that when the brightness reaches a certain value, the collected image reaches a stable state without significant changes.

Where $C_1$ and $C_2$ are constants, $I_{vis}$ represents the input visible source image, and $I_{vise}$ represents the enhanced result of visible image.

Dividing the grayscale values from 0 to 255 into 20 grayscales, the initial illumination brightness $l_0$ is the lowest brightness of the light, the remaining five brightness stages are set as ($l_1, l_2, l_3, l_4, l_5$), by collecting the gray value of the same color level displayed under different illumination intensity conditions, the variation of color levels under different illumination conditions is studied, the constants $c_{x1}$ and $c_{x2}$ in the power function at different brightness, in order to reduce the error and noise effect, calculate the average illumination brightness $l_{average}$ to obtain the best enhancement effect, the final constants $C_{average1}$ and $C_{average2}$ are as follows:

$$C_{average1} = \frac{\sum_{x=0}^{l} c_{x1}}{l}, l = 5$$

$$C_{average2} = \frac{\sum_{x=0}^{l} c_{x2}}{l}, l = 5 \tag{9}$$

Take the color block with gray value of 0, 20, 40 as an example, and the results acquired at different brightness stages are shown in Fig 2.

According to the fitting results, $C_1 = 16.176$ and $C_2 = 0.5339$ are obtained, and the final form of power function of simulated illumination enhancement is as follows:

$$I_{vise} = 16.176 * I_{vis}^{0.5339} \tag{10}$$

## 3.2 Infrared image enhancement base on adaptive histogram partition and brightness correction

The fusion of traditional infrared and visible images algorithms often only decomposes the image and operates on images of different frequency bands without paying attention to the image itself. Infrared images contain a lot of thermal radiation information, which should be widely concerned. The main purpose of infrared image enhancement is to enhance the edge sharpness of the image and improve the fuzzy state of the infrared image. It is more flexible and realistic to stretch the gray level in the adaptive histogram division. The brightness

correction uses the moderate gray level of the infrared image, the gray level is more rich, and the expression is enhanced.

In this paper, an algorithm combining AHP and BC [33] is used to enhance infrared images. At first, we use Gaussian Filtering(GF) and locally weighted scatter plot smoothing [34] to perform on the gray histogram of the original image:

$$p(I_k) = \sum_{j=k-w_1}^{k+w_1} p(I_j) \cdot k(j), k = w_1, w_1 + 1, \ldots L - w_1 - 1 \tag{11}$$

Where $p(k)$ refers to the smoothed Probability Distribution Functon(PDF) after GF. $1 * (2w_1 + 1)$ is the size of local window, in a 8-bit image $L = 256$. One-dimensional Gaussian kernel $k(\cdot)$ is defined as:

$$k(x) = \frac{1}{\sqrt{2\pi}\sigma} e^{-\frac{(x-x_0)^2}{2\sigma^2}} \tag{12}$$

Where $\sigma$ is a constant parameter deciding the weight of each neighboring PDF affecting the output, and current grayscale $I_k$ is represents $x_0 = k$.

Then partition the input image into two parts as: foreground $I_{ir}^f$ and background $I_{ir}^b$. Thus, the final enchanced image $I_{ire}$ can be obtained as:

$$I_{ire} = \{I_{ire}(x, y)\} = I_{ir}^f \cup I_{ir}^b \tag{13}$$

Where $I_{ir}^f$ and $I_{ir}^b$ can calculate by:

$$I_{ir}^f(I_k) = \sum_{t=1}^{j-1} R_t + R_j \cdot \frac{\aleph(I_k)}{C_j^\aleph}, j = k + w_1 \tag{14}$$

$$I_{ir}^b(I_k) = \sum_{j=1}^{i-1} R_t + R_i \cdot \frac{I_k - m_i}{m_{i+1} - m_i}, i = 6 \tag{15}$$

$R_x$ is metric to decide the re-mappde range, $\aleph(I_k)$ is the local contrast weighted distribution, $C_j^\aleph$ is the cumulative local contranst weighted distribution of $[m_j, m_{j+1}]$.

Finally, the formulation of the objective function can be deduced by formula Eq (14) as:

$$F = |M_o(m_1') - M_R| \tag{16}$$

Where $M_o(m_1')$ and $M_R$ stand for the mean of $I_{ir}$ and reference image $I_R$. $I_R$ just a image which mean brightness is suitable for hunman vision system. It is utilized to generate a standard mean intensity value $M_R$ to optimize the objective fuction value $F$.

## 3.3 Enhanced image fusion method based on K-L transform

For simplicity, let us take $I_{vis}(x, y)$ and $I_{vise}(x, y)$ as the input. Arrange these image as column vectors of a matrix $\bar{X}$. Make each row as an observation and each column as a variable to find the covariance matrix $C_{XY}$ of $\bar{X}$. Calculate eigen values $\sigma_1, \sigma_2$ and eigen vectors $\xi_1 = \begin{bmatrix} \xi_1 & (1) \\ \xi_1 & (2) \end{bmatrix}$ and $\xi_1 = \begin{bmatrix} \xi_2 & (1) \\ \xi_2 & (2) \end{bmatrix}$ of $C_{XY}$. The values of uncorrelated components $KL_1$

and $KL_2$ can giving by $\sigma_{\max}$:

$$\sigma_{\max} = \max(\sigma_1, \sigma_2) \tag{17}$$

$$KL_1 = \frac{\sigma_{\max}(1)}{\sum\limits_{i=1}^{2}\sigma_{\max}(i)}, i = 2$$

$$KL_2 = \frac{\sigma_{\max}(2)}{\sum\limits_{i=1}^{2}\sigma_{\max}(i)}, i = 2 \tag{18}$$

The fused result $I_{visef}$ of K-L transform can be calculated by:

$$I_{visef} = KL_1 * I_{vis}(x, y) + KL_2 * I_{vise}(x, y) \tag{19}$$

The extension for input images can be calculated by:

$$I_{visefN} = \sum_{n=1}^{N} KL_n * I_{visn}(x, y) \tag{20}$$

In the same way, the fused result $I_{iref}$ is given by:

$$I_{iref} = KL_1{'} * I_{ir}(x, y) + KL_2{'} * I_{ire}(x, y) \tag{21}$$

Where $KL_1{'}$ and $KL_2{'}$ represent the uncorrelated components in $I_{ir}$ and $I_{ire}$.

## 3.4 Fusion of infrared and visible images

LP fusion is performed on each spatial frequency layer separately, so that different fusion operators can be used to highlight the features and details of specific frequency bands according to the characteristics and details of different frequency bands of different decomposition layers.

An important property of the LP is that it is a complete image representation: the steps used to construct the pyramid may be reversed to accurately merge the resulting new image, LP fusion results are as follows [35]:

$$I_{lp} = LP_0 + (Expand(LP_1 + Expand(LP_2 + \ldots + Expand(LP_{N_l})))) \tag{22}$$

Where $LP_l$ is the $l$-th level image decomposed from LP and *Expand* operator is the inverse of *Reduce* operator.

In the final fusion stage, in order to better retain detailed information, this paper superimpose the result of LP fusion with the detailed image obtained in the section 2. The final fusion results are defined as follows:

$$I_{result} = I_{lp} + I_{ird} + I_{visd} \tag{23}$$

For the overall fusion process, it can be described in Algorithm 1.

**Algorithm 1**

```
Input: visible image I_vis, Infrared image I_ir.
  step 1. The detail layers I_ird and I_visd of infrared and visible image
can be obtain by Eqs 1-7.
  step 2. The visible enhanced image I_vise is obtained by Eqs 8-10. The
infrared enhanced image I_ire is obtained by Eqs 11-16.
```

```
    step 3. The K-L transform is performed on the source image and the
enhanced image to obtain the enhanced fusion image I_visef and I_iref by
Eqs 17-21.
    step 4. I_visef and I_iref were fused by LP to obtain I_lp by Eq 22.
    step 5. Calculate the final fused image I_result by Eq 23.
Output: Fused image I_result
```

## 4 Experimental results and analysis

All experiments in this paper are established on Matlab R2018a platform under Windows 10 operating system. To ensure the reliability of the experimental effect, the infrared image and visible image data selected are all from TNO_Image_Fusion_Dataset [36] and Li [37]. The purpose of the experiment is to verify the proposed method with objective and subjective standards and compare it with existing methods.

In this paper, 4 groups of infrared and visible images that have been commonly used are selected as the test images, named "Natocamp", "Nightstreet", "Kaptein", and "Gun", respectively. The original registration test image is shown in Fig 3.

### 4.1 Comparison method and experimental parameter setting

Experimental parameter settings are shown in Table 1.

### 4.2 Filtering algorithm

It is very important to pre-process the source image and separate the detail layer. The quality of the detail layer directly determines the final fusion image's performance in detail texture. In the experiment, the mean value [38], minimum value [39], Gaussian value [40] and median value [41] and the anisotropic diffusion filtering method are used to separate the infrared image and visible image at the detail layer. The detail separation layer results of Nato camp are shown in Fig 4.

### 4.3 Fitting method to determine the visible image enhancement algorithm

In this study, the experiment is combined with the real life. By implementing white light illumination of different brightness, the image of color level under the illumination condition is taken and collected by the camera, and the relationship between the variation of color level in low light and different illumination is explored, so as to propose a simple and practical

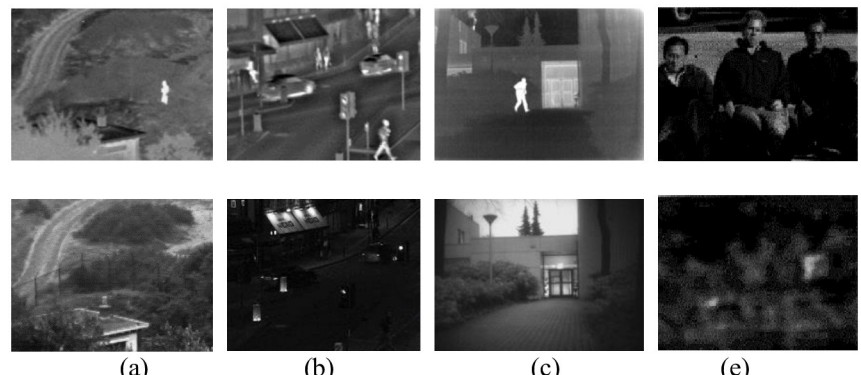

(a) (b) (c) (e)

**Fig 3. Original registration test image.**

**Table 1. Experimental parameter setting.**

| parameter | Meaning | Default value |
|---|---|---|
| $C_1$ | Enhancement coefficient of Power function | 16.176 |
| $C_2$ | Exponential percentage of Power function | 0.5339 |
| $w_1$ | Window radius of gaussian filter | 4 |
| $\sigma_1$ | Variance of gaussian filter | 0.7 |
| $w_2$ | Local minimum checks the length of the sliding window | 9 |
| $w_3$ | Length of the local entropy window | 7 |
| $\varepsilon_1$ | The lower bound of the weight function $\aleph(x)$ | 0.0001 |
| $\varepsilon_2$ | The upper bound of the weight function $\aleph(x)$ | 0.9999 |
| $t_{max}$ | The maximum iteration of particle swarm optimization | 10 |
| $N$ | The maximum iteration of particle swarm optimization | 10 |
| $c_1$ | Learning factor of particle swarm optimization algorithm | 0.5 |
| $c_2$ | Learning factor of particle swarm optimization algorithm | 0.5 |
| $\omega_{max}$ | The maximum weight of inertia for particle swarm optimization | 0.9 |
| $\omega_{min}$ | The minimum inertia weight of particle swarm optimization | 0.1 |
| $N_l$ | The number of Laplacian pyramid levels | 4 |

Where $C_1$ and $C_2$ are calculated for this paper, and the rest parameters are derived from Wan [32].

simulated lighting enhancement algorithm in dark scenes. The histogram of the chromatic scale image acquired under the brightness is shown in Fig 5.

It can be found that the color block with low gray value has two peaks in its histogram, because there is a white area around the color block, that is, the gray value under the current brightness of the color block with another peak of gray value of gray 255 similarly, the grayscale values of the remaining five chrominance level images are collected. The changes of gray

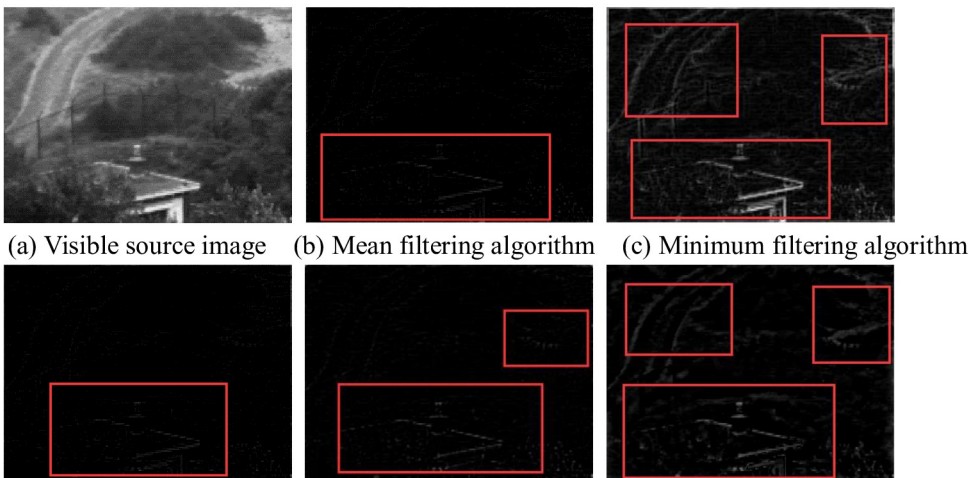

(a) Visible source image (b) Mean filtering algorithm (c) Minimum filtering algorithm

(d) Gaussian filtering algorithm (e) Median filtering (f) Anisotropic diffusion filtering

**Fig 4. Detail separation layer results of Nato camp.** In Fig 4, (b) and (d) can only get some detailed features, such as the outline of the building. However, the rest of the details are not reflected, such as roads and plants. the result of (c) is good, but contains a lot information do not belong to the background. The building of (e) can get a small amount of road information. The filtering algorithm (f) selected in this paper has rich details and clear texture, which is superior to the filtering decomposition results of other algorithms.

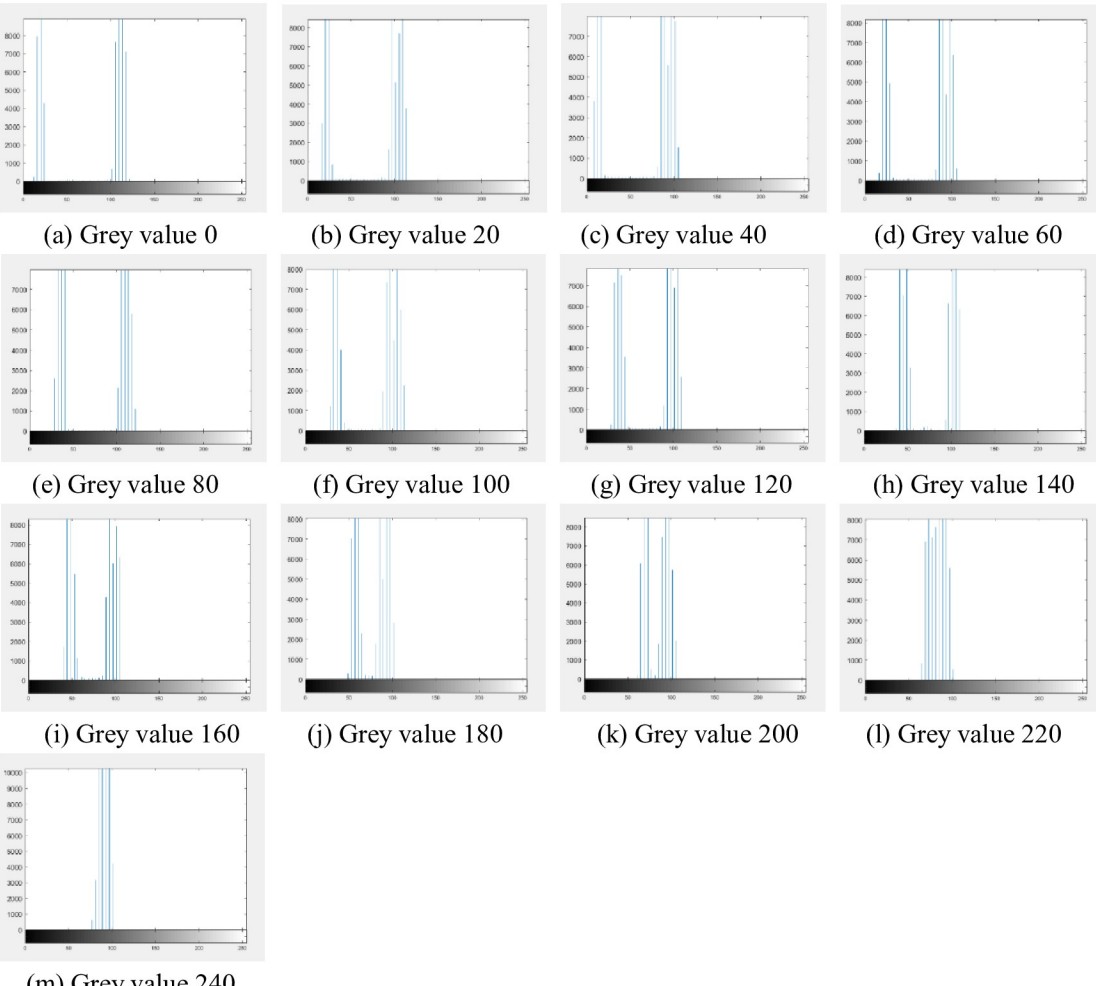

**Fig 5. Histogram of the gradation image collected at the lowest brightness.**

values corresponding to $l_0$ brightness to $l_1$ brightness, $l_0$ brightness to $l_2$ brightness, $l_0$ brightness to $l_3$ brightness, $l_0$ brightness to $l_4$ brightness and $l_0$ brightness to $l_5$ brightness are shown in Tables 2 to 6:

By observing the objective data, it can be known that when the illumination reaches a certain level, the gray value of the acquired chroma level will tend to be a stable one. In order to ensure the robustness of the algorithm, the average illumination $l_{average}$ is calculated, and the grayscale values changes corresponding to $l_0$ brightness to $l_{average}$ brightness is shown in Table 7.

**Table 2. Correspondence of grayscale values between brightness $l_0$ and brightness $l_1$.**

| Brightness | 0 | 20 | 40 | 60 | 80 | 100 | 120 | 140 | 160 | 180 | 200 | 220 | 240 | 255 |
|---|---|---|---|---|---|---|---|---|---|---|---|---|---|---|
| $l_0$ | 14 | 20 | 22 | 26 | 36 | 34 | 36 | 44 | 46 | 58 | 70 | 82 | 91 | 100 |
| $l_1$ | 46 | 60 | 68 | 76 | 81 | 93 | 93 | 105 | 121 | 129 | 139 | 143 | 165 | 185 |

**Table 3. Correspondence of grayscale values between brightness $l_0$ and brightness $l_2$.**

| Brightness | 0 | 20 | 40 | 60 | 80 | 100 | 120 | 140 | 160 | 180 | 200 | 220 | 240 | 255 |
|---|---|---|---|---|---|---|---|---|---|---|---|---|---|---|
| $l_0$ | 14 | 20 | 22 | 26 | 36 | 34 | 36 | 44 | 46 | 58 | 70 | 82 | 91 | 100 |
| $l_2$ | 62 | 80 | 85 | 95 | 105 | 113 | 117 | 133 | 137 | 147 | 155 | 153 | 176 | 204 |

**Table 4. Correspondence of grayscale values between brightness $l_0$ and brightness $l_3$.**

| Brightness | 0 | 20 | 40 | 60 | 80 | 100 | 120 | 140 | 160 | 180 | 200 | 220 | 240 | 255 |
|---|---|---|---|---|---|---|---|---|---|---|---|---|---|---|
| $l_0$ | 14 | 20 | 22 | 26 | 36 | 34 | 36 | 44 | 46 | 58 | 70 | 82 | 91 | 100 |
| $l_2$ | 68 | 85 | 89 | 97 | 109 | 117 | 119 | 133 | 139 | 152 | 159 | 153 | 176 | 208 |

**Table 5. Correspondence of grayscale values between brightness $l_0$ and brightness $l_4$.**

| Brightness | 0 | 20 | 40 | 60 | 80 | 100 | 120 | 140 | 160 | 180 | 200 | 220 | 240 | 255 |
|---|---|---|---|---|---|---|---|---|---|---|---|---|---|---|
| $l_0$ | 14 | 20 | 22 | 26 | 36 | 34 | 36 | 44 | 46 | 58 | 70 | 82 | 91 | 100 |
| $l_2$ | 70 | 85 | 91 | 99 | 109 | 117 | 121 | 133 | 139 | 145 | 155 | 151 | 174 | 208 |

**Table 6. Correspondence of grayscale values between brightness $l_0$ and brightness $l_5$.**

| Brightness | 0 | 20 | 40 | 60 | 80 | 100 | 120 | 140 | 160 | 180 | 200 | 220 | 240 | 255 |
|---|---|---|---|---|---|---|---|---|---|---|---|---|---|---|
| $l_0$ | 14 | 20 | 22 | 26 | 36 | 34 | 36 | 44 | 46 | 58 | 70 | 82 | 91 | 100 |
| $l_2$ | 70 | 85 | 91 | 97 | 109 | 117 | 121 | 133 | 137 | 145 | 155 | 155 | 175 | 208 |

The data in Table 7 are fitted. The fitting function includes linear function, exponential function, logarithmic function and power function. The fitting results of each function are shown in Fig 6.

Goodness of Fit refers to the degree of fitting of regression lines to observed values. The statistical measure of Goodness of Fit is determined on coefficient $R^2$. $R^2$ has a maximum value of 1 and a minimum value of 0. The closer the value of $R^2$ is to 1, the better the fitting degree of the regression line to the observed value is. On the contrary, when the value of $R^2$ is close to 0, it indicates that the fitting degree of the regression line to the observed value is worse. From the perspective of objective index $R^2$, the index $AR^2$ of power function is the largest among the four functions, which proves that its goodness of fit is higher. The enhancement results of Nightstreet are shown in Fig 7.

The visible source image and the enhancement image are K-L transformed, and the fusion results of Nightstreet are shown in Fig 8.

## 4.4 Enhanced algorithm of infrared image

The infrared image processing in this study is based on an infrared image adaptive histogram Partition and brightness correction and enhancement algorithm proposed by Wan [30]. Wan

**Table 7. Correspondence of grayscale values between brightness $l_0$ and brightness $l_{average}$.**

| Brightness | 0 | 20 | 40 | 60 | 80 | 100 | 120 | 140 | 160 | 180 | 200 | 220 | 240 | 255 |
|---|---|---|---|---|---|---|---|---|---|---|---|---|---|---|
| $l_0$ | 14 | 20 | 22 | 26 | 36 | 34 | 36 | 44 | 46 | 58 | 70 | 82 | 91 | 100 |
| $l_2$ | 63.2 | 79 | 84.8 | 92.8 | 102.6 | 111.4 | 114.2 | 127.4 | 134.6 | 143.6 | 152.6 | 151 | 173.2 | 202.6 |

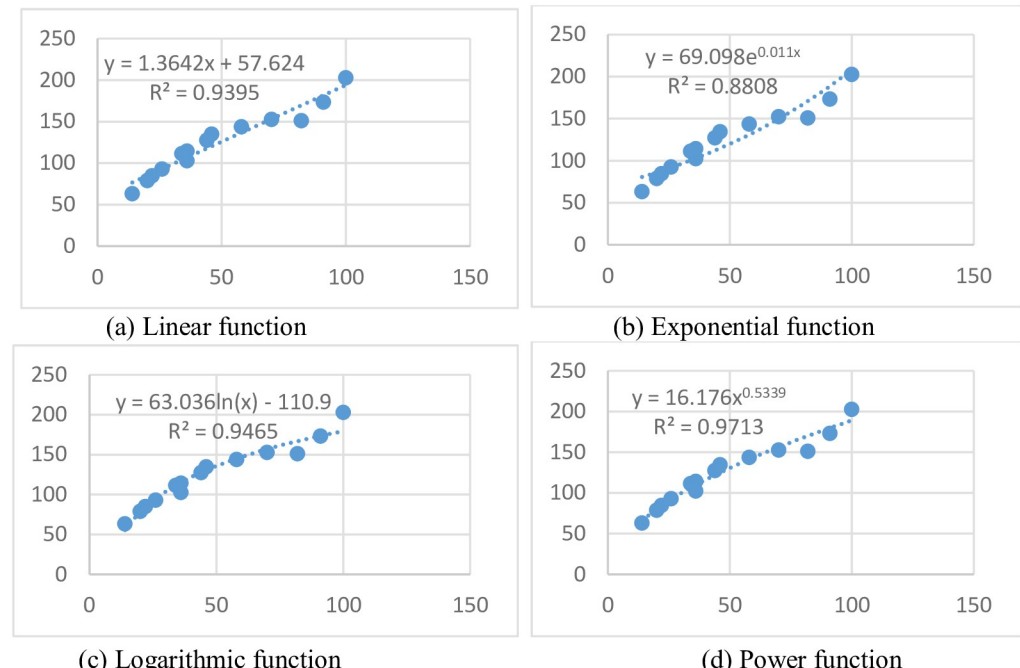

(a) Linear function

(b) Exponential function

(c) Logarithmic function

(d) Power function

**Fig 6. Fitting results of each function.**

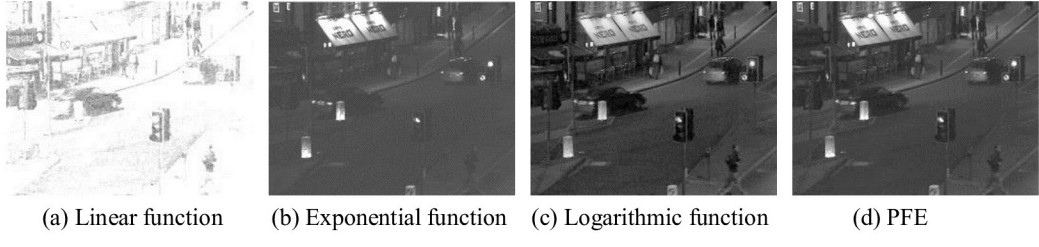

(a) Linear function (b) Exponential function (c) Logarithmic function (d) PFE

**Fig 7. Enhancement results of Nightstreet.** In Fig 7, (a) is excessively enhanced, resulting in overexposure and poor practicability. In (b), there is a problem similar to (a), which is excessively enhanced and in the loss of much information in the image. In (c), the effect is better, but there is more noise after enhancement. In (d), it achieves the best enhancement effect.

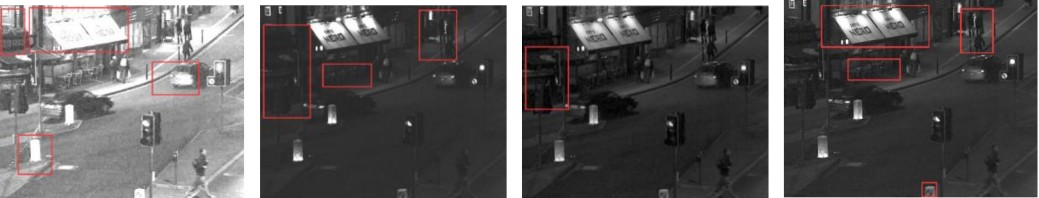

(a) Linear function fusion (b) Exponential function fusion (c) Exponential function fusion (d) K-L transform fusion

**Fig 8. Fusion results of Nightstreet.** In Fig 8, (a) is too bright that areas of high brightness overexposed, especially those marked by the red box. In (b) the brightness is low, and the area marked by red box is not conducive to observation. In (c), the fusion effect is good, but the marked corner area is not bright enough. In (d), it achieves the best fusion effect, complete image details, appropriate brightness, conducive to human eye observation.

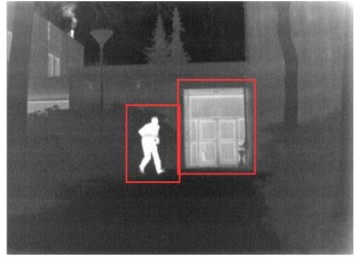 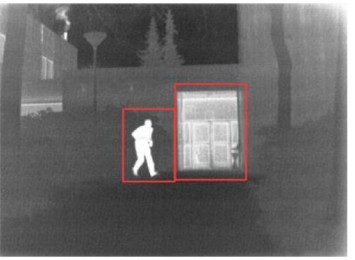 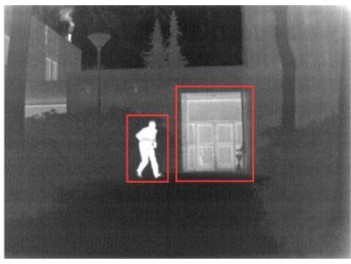

(a) The infrared source map  (b) Infrared enhancement result  (c) K-L transform fusion

**Fig 9. Enhancement and fusion results of Kaptein.**

proposed an infrared image adaptive histogram Partition and brightness correction and enhancement algorithm [33], and K-L transform of the results with the infrared source image to ensure that the original information of the source image is maintained while the infrared image is enhanced.

The enhancement and fusion results of Kaptein are shown in Fig 9.

## 4.5 Subjective evaluation

To better reflect the advantages of the proposed algorithm, enhanced infrared and visible images are used as new fusion source images, the experiment will test the proposed method and a variety of classical fusion algorithms in multiple fields: Gradient-Transform(GTF) algorithm [24], Pulse Coupled Neural Network(PCNN) algorithm [42], Dual Tree Complex Wavelet Transform(DTCWT) algorithm [43], CVT [17, 44] algorithm, Multi-resolution Singular Value Decomposition(MSVD) algorithm [45], Guided Filtering(GF) [46] algorithm, latent Low-Rank Representation(LRR) [47] algorithm, and a comparison between subjective and objective is conducted.

The fusion results of Natocamp are shown in Fig 10. In (a), the grass, the figure and the plants around the building are very blurred, with serious overall blurriness. In (b), the infrared

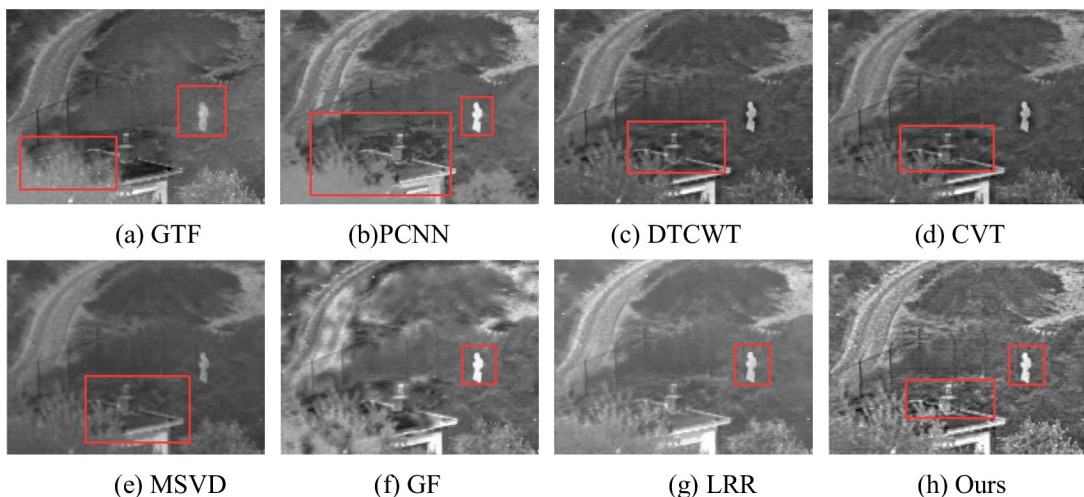

(a) GTF  (b)PCNN  (c) DTCWT  (d) CVT

(e) MSVD  (f) GF  (g) LRR  (h) Ours

**Fig 10. Fusion results of Natocamp.**

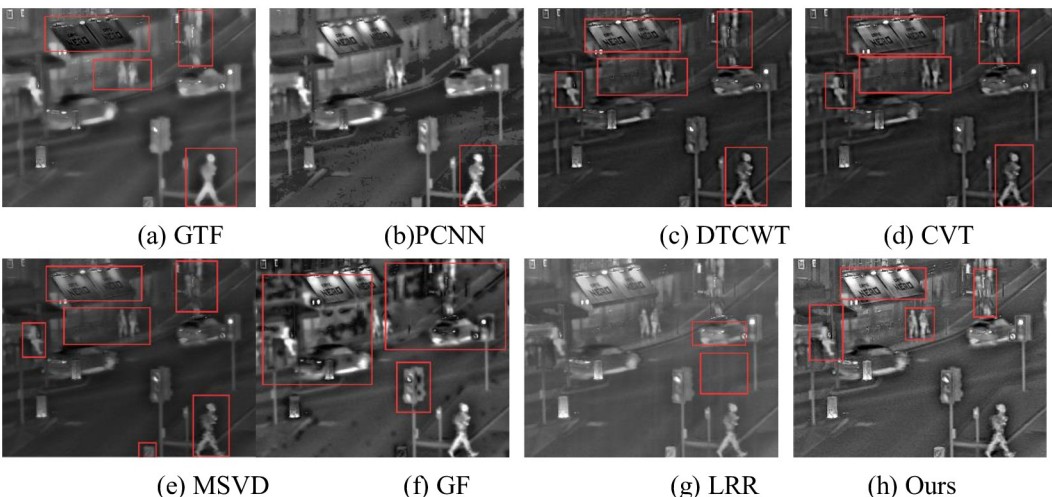

(a) GTF  (b)PCNN  (c) DTCWT  (d) CVT

(e) MSVD  (f) GF  (g) LRR  (h) Ours

**Fig 11. Fusion results of Nightstreet.**

target "human" is very clear, but the "plant" has been blurred, and the fusion distortion appears near the "building". In (c) and (d), the area between the plant and the target is blurred. In (e) and (g), details are missing in the enclosure and plants are blurred. In (f), although there are prominent objectives, the background information is too vague to facilitate observation. In (h), roads, fences, plants, etc. are clearly defined and rich in detail.

The fusion results of Nightstreet are shown in Fig 11. In (a), the infrared target is relatively obvious, such as the tire of people and cars, but the effect is poor on the letters on signs, signs at the lower right corner, and the display of pedestrians at the upper right corner. In (b), there are many inconsistent fusion and artifact. In(c) and (d), the letters on the signs are better displayed, but the pedestrian and pavement outside the shops are more blurred. In (e), the overall picture is dark, with details such as letters and signs in the lower right corner are blurred. In (f), infrared targets are obvious, but many black spots are generated after fusion. In (g), the picture is generally good but lacking in details such as road and seats outside shops. In (h), the infrared target is clear, the lighting effect is good, the road and other details are complete.

The fusion results of Kaptein are shown in Fig 12. In (a), the trees above the building produce artifacts, the two sides of the building are seriously blurred, and the details of human feet are lost. In (b), the infrared target is obvious, but the fusion results in a large number of artifacts and detail misalignment. For example, the ground presents obvious shadows. In (c), the outline of the plants above the building is obvious, but there are a few artifacts on both sides of the plants and people. In (d), more artifacts are produced, such as ground and sky artifacts of varying degrees. In (e), the image is generally fuzzy and details of plants on both sides of the image are lost. In (g), the fused images are generally poor and lose their value for observation. In (f), the details of plants and people above the building are more complete, but the details of plants and ground on both sides of the road are missing. In (h), the details of people, rooms, roads and plants on either side are obvious and conducive to observation.

The fusion results of Gun are shown in Fig 13. In (a), the infrared target is obvious, but the information of face and background is lost and much artifact are produced. In (b), the fusion results in a "black block" on the face of the right character, and the hand and foot of the character cannot be recognized. In(c) to (e) infrared targets are relatively fuzzy, and the shadows in the background lose the gradual change characteristics and produce different degrees of artifact. In (f), the infrared target is obvious, but there are different degrees of "black" blocks on

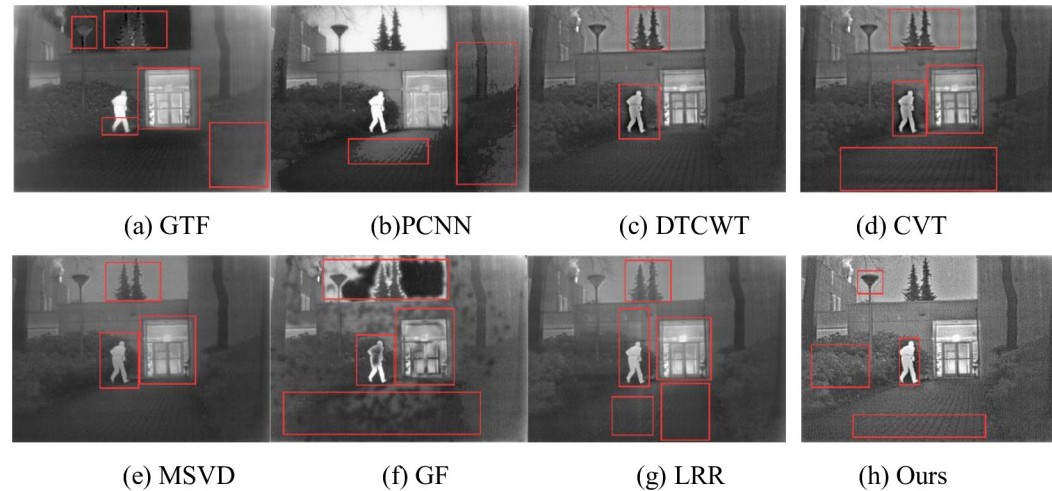

(a) GTF          (b)PCNN          (c) DTCWT          (d) CVT

(e) MSVD          (f) GF          (g) LRR          (h) Ours

**Fig 12. Fusion results of Kaptein.**

the left face and behind the other two people. In (g), only the infrared target of the gun is obvious, and the overall brightness is low. In (h), the infrared target is clear, and the texture after fusion is natura. For example, the shadow in the background can better reflect the real situation of the original scene, which is conducive to human observation.

## 4.6 Objective evaluation

Spatial Frequency(SF) [48], Mean Gradient(MG) [49], Energy Gradient(EG) [50], Edge Intensity(EI) [51], $Q^{AB/F}$ [52], $L^{AB/F}$ [53], Mutual Information(MI) [54], Structural Similarity (SSIM) [55] are selected as objective evaluation indexes to evaluate the fusion results. SF reflects the overall activity of the image in the spatial domain. MG reflects the contrast of image expressiveness from large to small details, which can be used to evaluate the degree of image blurring. EG is used to evaluate the clarity of fused images. EI examines the grayscale change of each pixel in a certain field to reflect the intensity of the edge contours of the fused image. $Q^{AB/F}$ is the important of evaluating the success of gradient information transfer from the inputs to the fused image. $L^{AB/F}$ is a measure of the information lost during the fusion

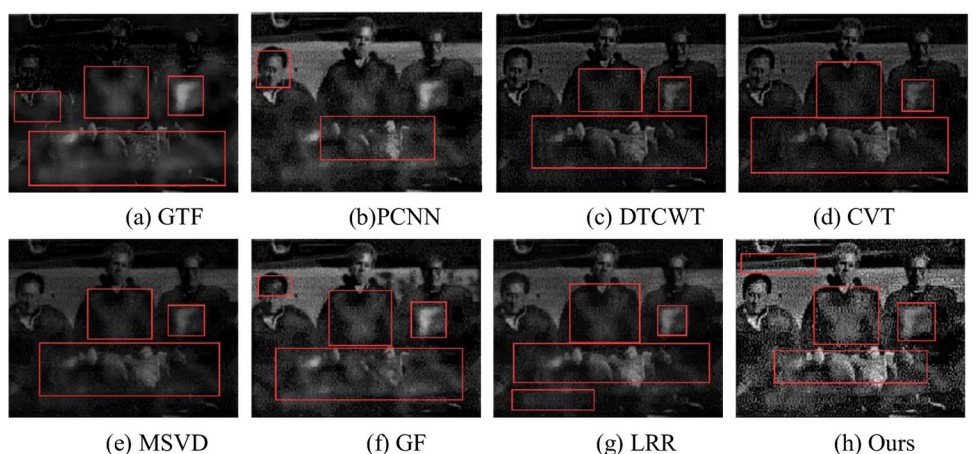

(a) GTF          (b)PCNN          (c) DTCWT          (d) CVT

(e) MSVD          (f) GF          (g) LRR          (h) Ours

**Fig 13. Fusion results of Gun.**

**Table 8. SF quality assessment results.**

| Method | Nato camp | Nightstreet | Kaptein | Gun | Rank |
|---|---|---|---|---|---|
| GTF | 9.4134 | 10.1375 | 7.4399 | 13.4092 | 2 |
| PCNN | 10.9371 | 12.4761 | 9.8931 | 13.7352 | 4 |
| DTCWT | 12.0704 | 13.4899 | 9.3758 | 20.6502 | 6 |
| CVT | 12.1543 | 13.4617 | 9.5279 | 19.7155 | 5 |
| MSVD | 9.3133 | 10.9151 | 8.3924 | 16.4483 | 3 |
| GF | 12.7966 | 15.273 | 9.3249 | 20.1855 | 7 |
| LRR | 9.5494 | 9.946 | 6.9366 | 13.4682 | 1 |
| Ours | 22.4171 | 21.0435 | 17.5279 | 24.4952 | **8** |

**Table 9. MG quality assessment results.**

| Method | Nato camp | Nightstreet | Kaptein | Gun | Rank |
|---|---|---|---|---|---|
| GTF | 3.4104 | 2.4483 | 2.4674 | 4.3272 | 2 |
| PCNN | 3.9276 | 3.8441 | 3.4399 | 4.9695 | 4 |
| DTCWT | 4.1714 | 3.2442 | 3.5085 | 8.1831 | 5 |
| CVT | 4.3341 | 3.2681 | 3.6844 | 8.1622 | 6 |
| MSVD | 2.8278 | 2.1762 | 2.5422 | 5.3437 | 3 |
| GF | 4.7413 | 4.509 | 3.6772 | 7.9175 | 7 |
| LRR | 3.4564 | 2.4135 | 2.529 | 3.8639 | 1 |
| Ours | 8.2212 | 5.6779 | 7.0615 | 10.813 | **8** |

process. MI represents the amount of information the fused image obtains from the input images. SSIM reflects the structural similarity between the input images and the fused image. The quality assessment results are shown from Tables 8 to 15. We rank the 8 algorithms in order from high to low according to the results of performance indicators.

The higher the objective index SF, MG, EG, EI, $Q^{AB/F}$, MI, and SSIM the better the fusion effect will be, while the $L^{AB/F}$ is the opposite. Combined with the data in Tables 8 to 11 and 13, it can be seen that in the performance of its own image, the algorithm in this paper performs well, has good edge contour characteristics, and the loss of fusion is the least. However, the connection with the source image is slightly weaker. The reason is that after Laplacian fusion, the features and details from different images may be merged, resulting in a change in the feature structure.

**Table 10. EG quality assessment results.**

| Method | Nato camp | Nightstreet | Kaptein | Gun | Rank |
|---|---|---|---|---|---|
| GTF | 4.1171 | 2.8751 | 3.4671 | 4.7224 | 1 |
| PCNN | 4.7234 | 4.7668 | 4.1894 | 5.3862 | 4 |
| DTCWT | 5.1685 | 4.2148 | 4.5475 | 8.9315 | 5 |
| CVT | 5.2749 | 4.2272 | 4.7142 | 8.9381 | 6 |
| MSVD | 4.3053 | 3.2597 | 4.1644 | 6.3486 | 3 |
| GF | 5.6218 | 5.4833 | 4.4204 | 8.6822 | 7 |
| LRR | 4.4571 | 3.2178 | 3.2708 | 6.2593 | 2 |
| Ours | 10.6692 | 7.809 | 9.25 | 11.8409 | **8** |

**Table 11. EI quality assessment results.**

| Method | Nato camp | Nightstreet | Kaptein | Gun | Rank |
|---|---|---|---|---|---|
| GTF | 3.57E+01 | 2.59E+01 | 2.36E+01 | 4.68E+01 | 2 |
| PCNN | 4.10E+01 | 4.05E+01 | 3.51E+01 | 5.27E+01 | 4 |
| DTCWT | 4.36E+01 | 3.42E+01 | 3.47E+01 | 8.67E+01 | 5 |
| CVT | 4.54E+01 | 3.45E+01 | 3.63E+01 | 8.68E+01 | 6 |
| MSVD | 2.81E+01 | 2.24E+01 | 2.26E+01 | 5.60E+01 | 1 |
| GF | 5.02E+01 | 4.79E+01 | 3.76E+01 | 8.33E+01 | 7 |
| LRR | 3.85E+01 | 2.71E+01 | 2.48E+01 | 6.13E+01 | 3 |
| Ours | 8.41E+01 | 5.76E+01 | 6.78E+01 | 1.1496 e+02 | **8** |

**Table 12. $Q^{AB/F}$ quality assessment results.**

| Method | Nato camp | Nightstreet | Kaptein | Gun | Rank |
|---|---|---|---|---|---|
| GTF | 0.6212 | 0.6709 | 0.5835 | 0.5131 | 3 |
| PCNN | 0.5945 | 0.5732 | 0.6386 | 0.5699 | 2 |
| DTCWT | 0.7360 | 0.8018 | 0.7951 | 0.8558 | 8 |
| CVT | 0.7178 | 0.7969 | 0.7859 | 0.8458 | 7 |
| MSVD | 0.5167 | 0.5534 | 0.6128 | 0.6921 | 1 |
| GF | 0.7372 | 0.8380 | 0.7266 | 0.8398 | 6 |
| LRR | 0.6980 | 0.7430 | 0.6738 | 0.7355 | 4 |
| Ours | 0.7190 | 0.8432 | 0.7000 | 0.8477 | **5** |

**Table 13. $L^{AB/F}$ quality assessment results.**

| Method | Nato camp | Nightstreet | Kaptein | Gun | Rank |
|---|---|---|---|---|---|
| GTF | 0.3687 | 0.3219 | 0.4080 | 0.4819 | 2 |
| PCNN | 0.3714 | 0.59 | 0.3012 | 0.4190 | 4 |
| DTCWT | 0.2424 | 0.1808 | 0.1796 | 0.1195 | 7 |
| CVT | 0.2512 | 0.1835 | 0.1793 | 0.1242 | 5 |
| MSVD | 0.4828 | 0.4425 | 0.3861 | 0.3043 | 1 |
| GF | 0.2184 | 0.1056 | 0.2211 | 0.1494 | 6 |
| LRR | 0.2965 | 0.2546 | 0.3239 | 0.2624 | 3 |
| Ours | 0.0783 | 0.0584 | 0.0584 | 0.0732 | **8** |

**Table 14. MI quality assessment results.**

| Method | Nato camp | Nightstreet | Kaptein | Gun | Rank |
|---|---|---|---|---|---|
| GTF | 1.4036 | 1.9337 | 2.0082 | 1.0713 | 6 |
| PCNN | 3.0891 | 3.0446 | 2.2695 | 2.4144 | 8 |
| DTCWT | 1.0248 | 1.1848 | 1.242 | 1.0047 | 3 |
| CVT | 0.9699 | 1.0867 | 1.161 | 0.9885 | 2 |
| MSVD | 1.0617 | 1.561 | 1.41 | 1.377 | 5 |
| GF | 1.3601 | 1.2918 | 1.9784 | 1.8904 | 7 |
| LRR | 1.1611 | 1.2243 | 1.3596 | 1.2153 | 4 |
| Ours | 0.9068 | 1.0387 | 0.9719 | 0.9485 | **1** |

**Table 15. SSIM quality assessment results.**

| Method | Nato camp | Nightstreet | Kaptein | Gun | Rank |
|---|---|---|---|---|---|
| GTF | 0.687 | 0.7438 | 0.7327 | 0.3878 | 4 |
| PCNN | 0.6552 | 0.6104 | 0.6474 | 0.3964 | 1 |
| DTCWT | 0.6903 | 0.654 | 0.7646 | 0.4895 | 6 |
| CVT | 0.6894 | 0.6535 | 0.7578 | 0.4874 | 5 |
| MSVD | 0.7155 | 0.6786 | 0.7787 | 0.4999 | 7 |
| GF | 0.6399 | 0.6332 | 0.6745 | 0.5065 | 3 |
| LRR | 0.7521 | 0.7657 | 0.7974 | 0.5222 | 8 |
| Ours | 0.6074 | 0.6785 | 0.6128 | 0.4795 | **2** |

**Table 16. The run time of test image.**

| Method | Nato camp | Nightstreet | Kaptein | Gun | Average time |
|---|---|---|---|---|---|
| GTF | 4.3875 | 34.9197 | 24.7382 | 21.0364 | 21.27045 |
| PCNN | 69.4682 | 236.8848 | 229.5092 | 185.1 | 180.2405 |
| DTCWT | 3.8365 | 29.2232 | 20.2963 | 16.8266 | 17.5456 |
| CVT | 4.2232 | 30.1543 | 21.2005 | 17.6661 | 18.3110 |
| MSVD | 3.8458 | 29.2119 | 20.6864 | 16.8188 | 17.6407 |
| GF | 3.8079 | 29.3465 | 20.8178 | 16.9012 | 17.7183 |
| LRR | 31.1413 | 154.1903 | 133.5535 | 112.0761 | 107.7403 |
| Ours | 3.676 | 28.7858 | 19.8234 | 16.4355 | **17.1801** |

The run time is also an important standard to evaluate the quality of this algorithm. The run time of test image are shown from Table 16.

According to Table 16, it can be seen that the running time of the algorithm in this paper is obviously less than that of other algorithms, so the time cost of this algorithm is the lowest. PCNN algorithm takes the longest time, the time cost of this algorithm is the highest.

## 5 Conclusion

On the one hand, the effect of image fusion depends on the quality of fusion algorithm, on the other hand, it also depends on the quality of source image. In order to better combine the characteristics of visible and infrared images and obtain more texture information, this paper proposes a new image enhancement fusion method combining K-L transform and LP fusion. Firstly, the anisotropic diffusion is used to extract the detail layer of the source image. According to the characteristics of the visible and infrared images, we select different enhancement algorithm. The power function enhancement algorithm is used to simulate the illumination of visible image to improve the brightness of the image and mine the details of the dark image. The infrared image is enhanced by adaptive histogram partition and brightness correction to highlight the characteristics of the target. Secondly, K-L transformation is performed between the enhanced images and the source image to form a new visible and infrared images to ensure that the image is enhanced without distortion and reduce artifacts. Finally, LP fusion is performed on the new visible and infrared images, and then the detail layer image is superimposed to obtain the fused image. The experimental results show that the method is subjectively clear texture, high visibility and good observability. In terms of objective indicators, the indicators of the image itself perform well, but the connection with the source image becomes weak. We will conduct further research on this issue in the later period to solve this problem.

## Author Contributions

**Conceptualization:** Hui Huang, Linlu Dong, Xiaofang Liu.

**Data curation:** Hui Huang, Zhishuang Xue, Xiaofang Liu, Caijian Hua.

**Formal analysis:** Hui Huang.

**Methodology:** Hui Huang.

**Project administration:** Hui Huang.

**Validation:** Hui Huang.

**Writing – original draft:** Hui Huang.

**Writing – review & editing:** Xiaofang Liu.

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
