## [Decision Letter · Decision Letter 0]

17 Nov 2020

PONE-D-20-32911

Fusion algorithm of visible and infrared image based on anisotropic diffusion and image enhancement

PLOS ONE

Dear Dr. Liu,

Thank you for submitting your manuscript to PLOS ONE. After careful consideration, we feel that it has merit but does not fully meet PLOS ONE’s publication criteria as it currently stands. Therefore, we invite you to submit a revised version of the manuscript that addresses the points raised during the review process.

I found this manuscript well written and interesting. As you will infer from below that there was a disagreement between the reviewers regarding enthusiasm for this work. Reviewer 1 recommended your work with minor revision. Reviewer 2  was of the opinion that proposed research did not describe a technically sound piece of research and recommended reject while the Reviewer 3 had made certain observations regarding your work and recommended major revision.

After thorough consideration of comments from all the reviewers, I felt that your study has merit but identified points that need to be addressed. Therefore, my decision is “major revision”.

Please revise the paper by incorporating all reviewer’s comments.

We look forward to receiving your revised manuscript.

Kind regards,

Gulistan Raja

Academic Editor

PLOS ONE

Journal Requirements:

Reviewers' comments:

Reviewer's Responses to Questions

**Comments to the Author**

1. Is the manuscript technically sound, and do the data support the conclusions?

Reviewer #1: Yes

Reviewer #2: No

Reviewer #3: Yes

2. Has the statistical analysis been performed appropriately and rigorously? 

Reviewer #1: Yes

Reviewer #2: Yes

Reviewer #3: Yes

3. Have the authors made all data underlying the findings in their manuscript fully available?

Reviewer #1: Yes

Reviewer #2: No

Reviewer #3: Yes

4. Is the manuscript presented in an intelligible fashion and written in standard English?

Reviewer #1: Yes

Reviewer #2: No

Reviewer #3: Yes

5. Review Comments to the Author

Reviewer #1: Fusion algorithm of visible and infrared image based on anisotropic diffusion and 2 image enhancement

Comments-

1) Compute Q AB/F, N AB/F and L AB/F and deduce the experimental interpretation. Gradient based metrics are important to validate your technique.

2) Use “gun’ Dataset. Gun dataset is captured in night mode contaminated with blur and noise. It is very important how your technique behave in that scenario. Compute proposed and other techniques on “Gun “Dataset.

3) Redraw figure 1 with smart art or m s visio . This figure seems to be very generic and simple. Kindly draw a more meaningful block diagram

4) Authors have missed to cite the popular and important review articles. There are 4 to 5 important review articles in literature in the field of image fusion. Please cite.

5) Visual interpretation of results are not very informative. Kindly provide the fine detailing how your architecture is better in preserving the homogeneous and non-homogeneous structures.

6) Abstract and conclusion are not up to mark. Please re – write.

7) Provide mathematics behind the proposed architecture. (If feasible only then).

8) There are few linguistic and grammatical errors. Please correct.

Reviewer #2: In this paper, authors proposed a new image fusion algorithm for visible and infrared images based on anisotropic diffusion and image enhancement.

1) In my view, the novelty of this work is minimum to be accepted by one of the the top journals such as PLOS ONE.

2) This method is almost similar to the work [24]. The main difference is: authors enhanced the contrast of the visible images as a preprocessing step before the fusion.

3) There are many grammatical errors scattered through out the paper which create the sloppy impression on the paper. Example, section 3.2 heading (3.2 Infrared image fusion base on ddaptive histogram partition and

178 brightness correction)

Reviewer #3: The paper presents a visible and infrared image fusion algorithm by anisotropic diffusion and image enhancement. The authors organize and implement extensive experiments to verify the effectiveness of the proposed method. Overall, the paper is presented clearly, and contains a decent amount of technical details. However, there are several major concerns after the reviewer’s study.

1. Considering the organization of the manuscript, the original proposal is not clearly evidenced. For example, the paper uses Adaptive Histogram Partition (AHP) and Brightness Correction (BC) algorithms to enhance infrared images to highlight the target object. As we know, the infrared images usually consist of thermal radiation characteristics. Please explain the necessity of enhancement of infrared images.

2. Edge gradient operator (QAB/F), mutual information and structural similarity (SSIM) are also two important metrics, please add them.

3. In experimental results, I think the brightness and detail of target are not satisfactory.

4.Some of the latest literatures should be cited. Infrared and visible images fusion using visual saliency

and optimized spiking cortical model in non-subsampled shearlet transform domain,Multimed Tools Appl (2019) 78:28609–28632.Infrared and visible image fusion based on convolutional neural network model and saliency detection via hybrid l0-l1 layer decomposition, J. Electron. Imaging 27(6), 063036 (2018).

Based on the comments above, the article lacks innovation. Reviewers cannot see a significant contribution to a journal paper, and there are still some problems in the manuscript that need to be optimized and supplemented

6. PLOS authors have the option to publish the peer review history of their article (what does this mean?). If published, this will include your full peer review and any attached files.

Reviewer #1: No

Reviewer #2: No

Reviewer #3: No

---

## [Author Response · Author response to Decision Letter 0]

8 Dec 2020

Dear Dr. Raja:

The article " Fusion algorithm of visible and infrared image based on anisotropic diffusion and image enhancement " has been revised in strict accordance with the comments of the review experts. The specific changes are as follows, please edit your review.

Comment 1：

1、Compute Q AB/F, N AB/F and L AB/F and deduce the experimental interpretation. Gradient based metrics are important to validate your technique.

Modify the description：

The article has added the Q AB/F, L AB/F, MI, and SSIM indicators in Table 12 to Table 15 according to the requirements. Table 16 is Rank score results.The details are as follows:

 [53-54] is the important of evaluating the success of gradient information transfer from the inputs to the fused image. is a measure of the information lost during the fusion process. Considering the correlation between and , we only calculate and here, but as a supplement, we add two other correlation functions MI[55] and SSIM[56]. MI represents the amount of information the fused image obtains from the input images. SSIM reflects the structural similarity between the input images and the fused image. The quality assessment results are shown from Table 8 to Table 15. We rank the 8 algorithms in order from from high to low according to the results of performance indicators.

Table 12. quality assessment results

Method Nato camp Nightstreet Kaptein Gun Rank score

GTF 0.3779 0.3768 0.3404 0.3145 2

PCNN 0.4282 0.5043 0.4504 0.5870 7

DTCWT 0.4319 0.4691 0.4620 0.5640 8

CVT 0.3885 0.4404 0.4271 0.5275 5

MSVD 0.2896 0.2929 0.3168 0.3811 1

GF 0.4518 0.4834 0.3911 0.6079 6

LRR 0.4109 0.4485 0.3812 0.4633 3

Ours 0.3929 0.4854 0.3588 0.5717 4

Table 13. quality assessment results

Method Nato camp Nightstreet Kaptein Gun Rank score

GTF 0.5827 0.5877 0.6211 0.6557 2

PCNN 0.3706 0.3415 0.3265 0.2666 7

DTCWT 0.4900 0.4503 0.4238 0.3126 5

CVT 0.4999 0.4646 0.4155 0.3274 4

MSVD 0.7047 0.6904 0.6662 0.6036 1

GF 0.4188 0.2652 0.4273 0.3414 6

LRR 0.5374 0.5208 0.5869 0.4979 3

Ours 0.1698 0.1638 0.1270 0.1616 8

Table 14.MI quality assessment results

Method Nato camp Nightstreet Kaptein Gun Rank score

GTF 1.4036 1.9337 2.0082 1.0713 8

PCNN 1.1279 1.2856 1.4142 1.3442 5

DTCWT 1.0248 1.1848 1.2420 1.0047 3

CVT 0.9699 1.0867 1.1610 0.9885 2

MSVD 1.0617 1.5610 1.4100 1.3770 6

GF 1.3601 1.2918 1.9784 1.8904 7

LRR 1.1611 1.2243 1.3596 1.2153 4

Ours 0.9068 1.0387 0.9719 0.9485 1

Table 15.SSIM quality assessment results

Method Nato camp Nightstreet Kaptein Gun Rank score

GTF 0.6870 0.7438 0.7327 0.3878 3

PCNN 0.7115 0.7478 0.7451 0.5163 7

DTCWT 0.6903 0.6540 0.7646 0.4895 5

CVT 0.6894 0.6535 0.7578 0.4874 4

MSVD 0.7155 0.6786 0.7787 0.4999 6

GF 0.6399 0.6332 0.6745 0.5065 2

LRR 0.7521 0.7657 0.7974 0.5222 8

Ours 0.6074 0.6785 0.6128 0.4795 1

The higher the objective index SF, MG, EG, EI, , MI, and SSIM the better the fusion effect will be, but is not. Combined with the data in Table 8 to Table 11 and Table 13, it can be seen that in the performance of its own image, the algorithm in this paper performs well, has good edge contour characteristics, and the loss of fusion is the least. However, the connection with the source image is slightly weaker. The reason is that after Laplacian fusion, the features and details from different images may be merged, resulting in a change in the feature structure.

[30] D Liu, D Zhou, R Nie, et al. “Infrared and visible image fusion based on convolutional neural network model and saliency detection via hybrid l0-l1 layer decomposition”. Journal of Electronic Imaging, 2018, 27(6): 063036.

[31] R Hou, R Nie, D Zhou, et al. “Infrared and visible images fusion using visual saliency and optimized spiking cortical model in non-subsampled shearlet transform domain”. Multimedia Tools and Applications, 2019, 78(20): 28609-28632.

[49] R Shapley, P Lennie. “Spatial frequency analysis in the visual system”. Annual review of neuroscience, 1985, 8(1): 547-581.

[50] B Pan, Z Lu, H Xie. “Mean intensity gradient: an effective global parameter for quality assessment of the speckle patterns used in digital image correlation”. Optics and Lasers in Engineering, 2010, 48(4): 469-477.

[51] J Gauss, JF Stanton, RJ Bartlett. “Coupled‐cluster open‐shell analytic gradients: Implementation of the direct product decomposition approach in energy gradient calculations”. The Journal of chemical physics, 1991, 95(4): 2623-2638.

[52] X Luo, Z Zhang, C Zhang, et al. “Multi-focus image fusion using HOSVD and edge intensity". Journal of Visual Communication and Image Representation, 2017, 45: 46-61.

[53] C Xydeas, V Petrovi. Objective Image Fusion Performance Measure, Electronics Letters, 2000, 36(4): 308-309.

[54] V Petrovic, C Xydeas. “Objective image fusion performance characterisation”. in:Tenth IEEE International Conference on Computer Vision. IEEE, 2005, 2: 1866-1871.

[55] M Seetha, IV MuraliKrishna, BL Deekshatulu. Data fusion performance analysis based on conventional and wavelet transform techniques. in: IEEE Proceedings on Geoscience and Remote Sensing Symposium, 2005, 4: 2842–2845.

[56] Z Wang, AC Bovik, HR Sheikh, et al. Image quality assessment: from errorvisibility to structural similarity. IEEE Transaction Image Process, 2004, 13(4):600–612.

Comment 2：

2、Use “gun’ Dataset. Gun dataset is captured in night mode contaminated with blur and noise. It is very important how your technique behave in that scenario. Compute proposed and other techniques on “Gun “Dataset.

Modify the description：

The article has added experiments on the 'gun' dataset according to the requirements. The details are as follows:

(a) GTF (b)PCNN (c) DTCWT (d) CVT

(e) MSVD (f) GF (g) LRR (h) Ours

Fig.14. Fusion results of Gun 

The fusion results of Gun are shown in Fig.13. In (a), the infrared target is obvious, but the information of face and background is lost and much artifact are produced. In (b), the fusion results in a "black block" on the face of the right character, and the hand and foot of the character cannot be recognized. In(c) to (e) infrared targets are relatively fuzzy, and the shadows in the background lose the gradual change characteristics and produce different degrees of artifact. In (f), the infrared target is obvious, but there are different degrees of "black" blocks on the left face and behind the other two people. In (g), only the infrared target of the gun is obvious, and the overall brightness is low. In (h), the infrared target is clear, and the texture after fusion is natura. For example, the shadow in the background can better reflect the real situation of the original scene, which is conducive to human observation.

Comment 3：

3、Redraw figure 1 with smart art or m s visio . This figure seems to be very generic and simple. Kindly draw a more meaningful block diagram.

Modify the description：

The article has revised the fig.1 by m s visio2013 according to the requirements, The details are as follows:

Fig.1. The structure of this paper

Comment 4：

4、Authors have missed to cite the popular and important review articles. There are 4 to 5 important review articles in literature in the field of image fusion. Please cite.

Modify the description：

The article has revised according to the requirements. The details are as follows:

[1] Y Zeng, W Huang, M Liu, et al. Fusion of satellite images in urban area: Assessing the quality of resulting images. in: 2010 18th International Conference on Geoinformatics. IEEE, 2010: 1-4.

[30] D Liu, D Zhou, R Nie, et al. “Infrared and visible image fusion based on convolutional neural network model and saliency detection via hybrid l0-l1 layer decomposition”. Journal of Electronic Imaging, 2018, 27(6): 063036.

[31] R Hou, R Nie, D Zhou, et al. “Infrared and visible images fusion using visual saliency and optimized spiking cortical model in non-subsampled shearlet transform domain”. Multimedia Tools and Applications, 2019, 78(20): 28609-28632.

[53] C Xydeas, V Petrovi. Objective Image Fusion Performance Measure, Electronics Letters, 2000, 36(4): 308-309.

[54] V Petrovic, C Xydeas. “Objective image fusion performance characterisation”. in:Tenth IEEE International Conference on Computer Vision. IEEE, 2005, 2: 1866-1871.

[55] M Seetha, IV MuraliKrishna, BL Deekshatulu. Data fusion performance analysis based on conventional and wavelet transform techniques. in: IEEE Proceedings on Geoscience and Remote Sensing Symposium, 2005, 4: 2842–2845.

[56] Z Wang, AC Bovik, HR Sheikh, et al. Image quality assessment: from errorvisibility to structural similarity. IEEE Transaction Image Process, 2004, 13(4):600–612.

Comment 5：

5、Visual interpretation of results are not very informative. Kindly provide the fine detailing how your architecture is better in preserving the homogeneous and non-homogeneous structures.

Modify the description：

In order to better preserve the homogeneous and non-homogeneous structures, the article has revised the part of the structure according to the requirements.The details are as follows:

(1) The detail layers and of infrared and visible images can be obtained by AD algorithm, to better preserve clear outline and detail information.

(2) We Propose a Power Function Enhancement (PFE) algorithm to simulate illumination for visible images to obtain visible enhanced images , and use Adaptive Histogram Partition (AHP) and Brightness Correction (BC) algorithms to enhance infrared images to obtain infrared enhanced images .

(3) Based on the K-L transform to fuse the enhanced image and the source image, the results and are used as new visible and infrared source image to fully retain the structural characteristics of the image and improve the fusion quality.

(4) In order to better preserve the characteristics of the image and improve the brightness of the image, Laplacian fusion is performed on and to obtain the fused image , and the weighted fusion of , and is used to obtain the result .

Comment 6：

6、Abstract and conclusion are not up to mark. Please re – write.

Modify the description：

The article has revised the introduction part according to the requirements. The abstract and conclusion are as follows:

Abstract: Aiming at the situation that the existing visible and infrared images fusion algorithms only focus on highlighting the infrared target, but cannot take into account the characteristics of infrared and visible images. Therefore, it is essential to preserve the detail of the source images and improve the quality of the source images, this paper proposes an image enhancement fusion algorithm combining Karhunen-Loeve transform and Laplacian pyramid fusion. The detail layer of the source images are obtained by anisotropic diffusion to get more abundant texture information. The infrared images adopt adaptive histogram division and brightness correction enhancement algorithm to highlight thermal radiation targets. A novel power function enhancement algorithm that simulates illumination is proposed for visible images to improve the contrast of visible images and facilitate human observation. In order to improve the fusion quality of images, the source images and the enhanced images are transformed by Karhunen-Loeve to form new visible and infrared images. Laplacian pyramid fusion is performed on the new visible and infrared images, and superimposed with the detail layer images to obtain the fusion result. Experimental results show that the method in this paper is superior to several representative image fusion algorithms in subjective visual effects on public data sets. In terms of objective evaluation, the fusion result performed well on the 8 evaluation indicators, and its own quality was high.

Conclusion:

The effect of image fusion depends on the quality of the fusion algorithm on the one hand, and the quality of the source image on the other. In order to better combine the characteristics of visible and infrared images and obtain more texture information, this paper proposes a new Image enhancement fusion method combining K-L transform and LP fusion. Firstly, the anisotropic diffusion is used to extract the detail layer of the source image. According to the characteristics of the visible and infrared images, we select different enhancement algorithm. The power function enhancement algorithm is used to simulate the illumination of visible image to improve the brightness of the image and mine the details of the dark image. The infrared image is enhanced by adaptive histogram partition and brightness correction to highlight the characteristics of the target. Secondly, K-L transformation is performed between the enhanced images and the source image to form a new visible and infrared images to ensure that the image is enhanced without distortion and reduce artifact. Finally, LP fusion is performed on the new visible and infrared images, and then the detail layer image is superimposed to obtain the fused image. The experimental results show that the method is subjectively clear texture, high visibility and good observability. In terms of objective indicators, the indicators of the image itself perform well, but the connection with the source image becomes weak. We will conduct further research on this issue in the later period to solve this problem.

Comment 7：

7、Provide mathematics behind the proposed architecture. (If feasible only then).

Modify the description：

The article has added more detailed attribute logic according to the requirements. The details are as follows:

In the third section, we provide the algorithm framework and the mathematical logic behind it. The flow of the algorithm in this paper is as follows:

Algorithm

Input: visible image , Infrared image .

Output: Fused image .

1. The detail layers and of infrared and visible image can be obtain by AD algorithm. Parameters 

2. Use the PFE for to get the , Parameters , . Use the AHP and BC for to get , Parameters , , , , , , , , , , , .

3. The K-L transform between the source image the enhanced image is performed to obtain the enhanced fusion image and .

4. and were fused by LP to obtain , Parameters .

5. Calculate the final fused image as .

Comment 8：

8、There are few linguistic and grammatical errors. Please correct.

Modify the description：

The article has been revised in English as required.

Comment 9：

9、In my view, the novelty of this work is minimum to be accepted by one of the the top journals such as PLOS ONE.

Modify the description：

The structure of the paper has been revised according to the requirements. The details are as follows:

Fig.1. The structure of this paper

Comment 10：

10、This method is almost similar to the work [24]. The main difference is: authors enhanced the contrast of the visible images as a preprocessing step before the fusion.

Modify the description：

The article has revised according to the requirements. The details are as follows:

I am glad you can read my article carefully. Based on your suggestion, I seriously on the framework of the article to be modified. Firstly, we use the anisotropic diffusion (AD) algorithm to obtain the detailed information of the source image. Second, we think that high quality source image is more conducive to image fusion, but because of the different characteristics of visible and infrared images, we use different enhancement algorithms for the two kinds of images. For visible image, we propose a simple and easy Power Function Enhancement (PFE) algorithm to simulate illumination, which is used to improve the brightness of the image and conducive to human observation. For infrared image, we use Adaptive Histogram Partition (AHP) and Brightness Correction (BC) algorithms. Third, in order to ensure the quality of fusion, the enhanced image and the source image for K-L transform. Fourth, we use the Laplacian pyramid to fuse new visible and infrared images to make the image fusion more coordinated in detail. Finally, superimpose the detail layer and the LP fusion result to obtain the final result. The flow of the algorithm in this paper is as follows:

Algorithm

Input: visible image , Infrared image .

Output: Fused image .

1. The detail layers and of infrared and visible image can be obtain by AD algorithm. Parameters 

2. Use the PFE for to get the , Parameters , . Use the AHP and BC for to get , Parameters , , , , , , , , , , , .

3. The K-L transform between the source image the enhanced image is performed to obtain the enhanced fusion image and .

4. and were fused by LP to obtain , Parameters .

5. Calculate the final fused image as .

Comment 11：

11、There are many grammatical errors scattered through out the paper which create the sloppy impression on the paper. Example, section 3.2 heading (3.2 Infrared image fusion base on ddaptive histogram partition and 178 brightness correction).

Modify the description：

The article has been revised in English as required.

Comment 12：

12、Considering the organization of the manuscript, the original proposal is not clearly evidenced. For example, the paper uses Adaptive Histogram Partition (AHP) and Brightness Correction (BC) algorithms to enhance infrared images to highlight the target object. As we know, the infrared images usually consist of thermal radiation characteristics. Please explain the necessity of enhancement of infrared images.

Modify the description：

Description of the necessity of infrared image enhancement. The details are as follows:

The main purpose of infrared image enhancement is to enhance the edge sharpness of the image and improve the fuzzy state of the infrared image. It is more flexible and realistic to stretch the gray level in the Adaptive Histogram Partition. The Brightness Correction uses the moderate gray level of the infrared image, the gray level is more rich, and the expression is enhanced. The results of infrared source image enhancement are as follows:

(a) (b) (c) (e)

It can be seen from figures (a) and (c) that the algorithm does not blindly improve the overall brightness of the image, resulting in over enhancement. In figure (b), the brightness of the image is significantly improved, and the infrared target is obvious. In figure (d), the fuzzy state of infrared image is obviously improved, and the gray level of infrared image is more convenient for observation.

Comment 13：

13、Edge gradient operator (Q AB/F), mutual information and structural similarity (SSIM) are also two important metrics, please add them.

Modify the description：

The article has added the , mutual information and SSIM indicators in Table 12, Table 15 and Table 16 according to the requirements, The details are as follows:

 is the important of evaluating the success of gradient information transfer from the inputs to the fused image. MI represents the amount of information the fused image obtains from the input images. SSIM reflects the structural similarity between the input images and the fused image.

Table 12. quality assessment results

Method Nato camp Nightstreet Kaptein Gun Rank

GTF 0.3779 0.3768 0.3404 0.3145 2

PCNN 0.4236 0.4327 0.3496 0.3348 3

DTCWT 0.4319 0.4691 0.4620 0.5640 7

CVT 0.3885 0.4404 0.4271 0.5275 5

MSVD 0.2896 0.2929 0.3168 0.3811 1

GF 0.4518 0.4834 0.3911 0.6079 8

LRR 0.4109 0.4485 0.3812 0.4633 4

Ours 0.3929 0.4854 0.3588 0.5717 5

Table 14.MI quality assessment results

Method Nato camp Nightstreet Kaptein Gun Rank

GTF 1.4036 1.9337 2.0082 1.0713 6

PCNN 3.0891 3.0446 2.2695 2.4144 8

DTCWT 1.0248 1.1848 1.2420 1.0047 3

CVT 0.9699 1.0867 1.1610 0.9885 2

MSVD 1.0617 1.5610 1.4100 1.3770 5

GF 1.3601 1.2918 1.9784 1.8904 7

LRR 1.1611 1.2243 1.3596 1.2153 4

Ours 0.9068 1.0387 0.9719 0.9485 1

Table 15.SSIM quality assessment results

Method Nato camp Nightstreet Kaptein Gun Rank score

GTF 0.6870 0.7438 0.7327 0.3878 4

PCNN 0.6552 0.6104 0.6474 0.3964 1

DTCWT 0.6903 0.6540 0.7646 0.4895 6

CVT 0.6894 0.6535 0.7578 0.4874 5

MSVD 0.7155 0.6786 0.7787 0.4999 7

GF 0.6399 0.6332 0.6745 0.5065 3

LRR 0.7521 0.7657 0.7974 0.5222 8

Ours 0.6074 0.6785 0.6128 0.4795 2

The performance of the algorithm in this paper on these three evaluation indicators is not as good as expected, because the LP fusion process will change the structure of the image. But in terms of the performance indicators of the image itself, its advantages are obvious, combined with visual experience, it is more suitable for human observation.

Comment 14：

1、In experimental results, I think the brightness and detail of target are not satisfactory.

Modify the description：

The article has made the some modifications in order to better retain the brightness and detail of target according to the requirements, The details are as follows:

In order to better highlight infrared targets and improve image brightness, LP fusion is performed on each spatial frequency layer separately, so that different fusion operators can be used to highlight the features and details of specific frequency bands according to the characteristics and details of different frequency bands of different decomposition layers.

An important property of the LP is that it is a complete image representation: the steps used to construct the pyramid may be reversed to accurately merge the resulting new image, LP fusion results are as follows[35]:

(22)

Where is the -th level image decomposed from LP. and operator is the inverse of operator.

In the final fusion stage, in order to better retain detailed information, Superimpose the result of LP fusion with the detailed image obtained in the section 2. The final fusion results are defined as follows:

(23)

Comment 15：

15、Some of the latest literatures should be cited. Infrared and visible images fusion using visual saliency and optimized spiking cortical model in non-subsampled shearlet transform domain,Multimed Tools Appl (2019) 78:28609–28632.Infrared and visible image fusion based on convolutional neural network model and saliency detection via hybrid l0-l1 layer decomposition, J. Electron. Imaging 27(6), 063036 (2018).

Modify the description：

The article has cited latest literatures according to the requirements. The details are as follows:

[30] D Liu, D Zhou, R Nie, et al. “Infrared and visible image fusion based on convolutional neural network model and saliency detection via hybrid l0-l1 layer decomposition”. Journal of Electronic Imaging, 2018, 27(6): 063036.

[31] R Hou, R Nie, D Zhou, et al. “Infrared and visible images fusion using visual saliency and optimized spiking cortical model in non-subsampled shearlet transform domain”. Multimedia Tools and Applications, 2019, 78(20): 28609-28632.

---

## [Decision Letter · Decision Letter 1]

22 Dec 2020

PONE-D-20-32911R1

Fusion algorithm of visible and infrared image based on anisotropic diffusion and image enhancement

PLOS ONE

Dear Dr. Liu,

Thank you for submitting your manuscript to PLOS ONE. After careful consideration, we feel that it has merit but does not fully meet PLOS ONE’s publication criteria as it currently stands. Therefore, we invite you to submit a revised version of the manuscript that addresses the points raised during the review process.

As you will infer from below that there was a disagreement regarding recommendation on your manuscript. Reviewer 3 was of the view that all comments have been addressed which he raised in last cycle of review and recommended accept. On the other hand, reviewer 1 had given some suggestions to further improve your work and  recommended minor revision. 

After considering comments of reviewer 1,  the editors decision is "minor revision".

Please incorporate comments raised by reviewer 1.

We look forward to receiving your revised manuscript.

Kind regards,

Gulistan Raja

Academic Editor

PLOS ONE

Reviewers' comments:

Reviewer's Responses to Questions

**Comments to the Author**

1. If the authors have adequately addressed your comments raised in a previous round of review and you feel that this manuscript is now acceptable for publication, you may indicate that here to bypass the “Comments to the Author” section, enter your conflict of interest statement in the “Confidential to Editor” section, and submit your "Accept" recommendation.

Reviewer #1: (No Response)

Reviewer #3: All comments have been addressed

2. Is the manuscript technically sound, and do the data support the conclusions?

Reviewer #1: (No Response)

Reviewer #3: Yes

3. Has the statistical analysis been performed appropriately and rigorously? 

Reviewer #1: (No Response)

Reviewer #3: Yes

4. Have the authors made all data underlying the findings in their manuscript fully available?

Reviewer #1: (No Response)

Reviewer #3: Yes

5. Is the manuscript presented in an intelligible fashion and written in standard English?

Reviewer #1: (No Response)

Reviewer #3: Yes

6. Review Comments to the Author

Reviewer #1: Fusion algorithm of visible and infrared image based on anisotropic diffusion and image enhancement

Comments-

Authors have incorporated all the suggested points. The appeal of the entire article is increased.

Author is suggested to verify the Q AB/F & L AB/F values for all tested and proposed dataset again.

Matlab codes Links –

1) https://www.mathworks.com/matlabcentral/fileexchange/43781-image-fusion-based-on-pixel-significance-using-cross-bilateral-filter?s_tid=srchtitle

2) https://www.mathworks.com/matlabcentral/fileexchange/43053-objective-fusion-performance-parameters-modified-fusion-artifacts-measure?s_tid=srchtitle

Reviewer #3: This work deals with image fusion. The subject is of interest to this journal, but I have some relevant considerations:

(1) Add a new sub-section to discuss the computational complexity. You are encouraged to add execution times to the tables.

(2) Add a qualitative table for integrating all simulated methods for different features.

7. PLOS authors have the option to publish the peer review history of their article (what does this mean?). If published, this will include your full peer review and any attached files.

Reviewer #1: No

Reviewer #3: **Yes: **Dongming Zhou

---

## [Author Response · Author response to Decision Letter 1]

30 Dec 2020

Dear Dr. Raja:

The article " Fusion algorithm of visible and infrared image based on anisotropic diffusion and image enhancement " has been revised in strict accordance with the comments of the review experts. The specific changes are as follows, please edit your review.

Comment 1：

Author is suggested to verify the Q AB/F & L AB/F values for all tested and proposed dataset again.

Modify the description：

According to the link provided by reviewer 1, this article has verified the values of Q AB/F, L AB/F of all tested and proposed dataset. The details are as follows:

Table 13. quality assessment results

Method Nato camp Nightstreet Kaptein Gun Rank score

GTF 0.6212 0.6709 0.5835 0.5131 3

PCNN 0.5945 0.5732 0.6386 0.5699 2

DTCWT 0.7360 0.8018 0.7951 0.8558 8

CVT 0.7178 0.7969 0.7859 0.8458 7

MSVD 0.5167 0.5534 0.6128 0.6921 1

GF 0.7372 0.8380 0.7266 0.8398 6

LRR 0.6980 0.7430 0.6738 0.7355 4

Ours 0.7190 0.8432 0.7000 0.8477 5

Table 14. quality assessment results

Method Nato camp Nightstreet Kaptein Gun Rank score

GTF 0.3687 0.3219 0.4080 0.4819 2

PCNN 0.3714 0.3759 0.3012 0.4190 3

DTCWT 0.2424 0.1808 0.1796 0.1195 6

CVT 0.2512 0.1835 0.1793 0.1242 5

MSVD 0.4828 0.4425 0.3861 0.3043 1

GF 0.2184 0.1056 0.2211 0.1494 7

LRR 0.2965 0.2546 0.3239 0.2624 4

Ours 0.0783 0.0584 0.0584 0.0732 8

The evaluation index used before in this paper is based on the reference [52] [53]. The codes Links –http://www.pudn.com/Download/item/id/2374577.html. We can calculate from the relationship between and . Although there are some slight changes in the numerical value of verification results, the overall performance of the algorithm is stable, and the ranking has not changed significantly.

Comment 2：

2、Add a new sub-section to discuss the computational complexity. You are encouraged to add execution times to the tables.

Modify the description：

The run time is one of the Objective evaluation, so add it into section 4.6. The details are as follows:

The run time is also an important standard to evaluate the quality of this algorithm. The run time of test image are shown from Table 17.

Table 17.The run time of test image

Method Nato camp Nightstreet Kaptein Gun average time rank

GTF 4.3875 34.9197 24.7382 21.0364 21.2704 3

PCNN 69.4682 236.8848 229.5092 185.1 180.2405 1

DTCWT 3.8365 29.2232 20.2963 16.8266 17.5456 7

CVT 4.2232 30.1543 21.2005 17.6661 18.3110 4

MSVD 3.8458 29.2119 20.6864 16.8188 17.6407 6

GF 3.8079 29.3465 20.8178 16.9012 17.7183 5

LRR 31.1413 154.1903 133.5535 112.0761 107.7403 2

Ours 3.676 28.7858 19.8234 16.4355 17.1801 8

According to table 16, it can be seen that the running time of the algorithm in this paper is obviously less than that of other algorithms, so the time cost of this algorithm is the lowest. PCNN algorithm takes the longest time, the time cost of this algorithm is the highest.

Comment 3：

3、Add a qualitative table for integrating all simulated methods for different features.

Modify the description：

The article has revised according to the requirements. Add a qualitative table in section 3.4. The details are as follows:

For the overall fusion process, it can be described in algorithm.

Algorithm

Input: visible image , Infrared image .

Step 1. The detail layers and of infrared and visible image can be obtain by Eq. (1)-(7). 

Step 2. The visible enhanced image is obtained by Eq. (8)-(10). The infrared enhanced image is obtained by Eq. (11)-(16).

Step 3. The K-L transform is performed on the source image and the enhanced image to obtain the enhanced fusion image and by Eq. (17)-(21).

Step 4. and were fused by LP to obtain by Eq. (22).

Step 5. Calculate the final fused image by Eq. (23).

Output: Fused image .

---

## [Decision Letter · Decision Letter 2]

4 Jan 2021

Fusion algorithm of visible and infrared image based on anisotropic diffusion and image enhancement

PONE-D-20-32911R2

Dear Dr. Liu,

We’re pleased to inform you that your manuscript has been judged scientifically suitable for publication and will be formally accepted for publication once it meets all outstanding technical requirements.

Kind regards,

Gulistan Raja

Academic Editor

PLOS ONE

Additional Editor Comments (optional):

Reviewers' comments:

Reviewer's Responses to Questions

**Comments to the Author**

1. If the authors have adequately addressed your comments raised in a previous round of review and you feel that this manuscript is now acceptable for publication, you may indicate that here to bypass the “Comments to the Author” section, enter your conflict of interest statement in the “Confidential to Editor” section, and submit your "Accept" recommendation.

Reviewer #1: (No Response)

Reviewer #3: All comments have been addressed

2. Is the manuscript technically sound, and do the data support the conclusions?

Reviewer #1: (No Response)

Reviewer #3: Yes

3. Has the statistical analysis been performed appropriately and rigorously? 

Reviewer #1: (No Response)

Reviewer #3: Yes

4. Have the authors made all data underlying the findings in their manuscript fully available?

Reviewer #1: (No Response)

Reviewer #3: Yes

5. Is the manuscript presented in an intelligible fashion and written in standard English?

Reviewer #1: (No Response)

Reviewer #3: Yes

6. Review Comments to the Author

Reviewer #1: All the edits are done properly, manuscript can be accepted.

It can be considered now without any further changes.

Reviewer #3: In this manuscript, the base layerche source image are obtained through anisotropic diffusion, the infrared image is processed with two kinds of operations, adaptive histogram Division and Brightness Correction, to highlight

the target object. In the dark scene image, the visible image is regarded as a lowilluminance image, and a power function enhancement algorithm is proposed to improve image brightness and magnify details. The source image and the enhanced image are fused to form new visible image and infrared image. Perform Karhunen-Loeve transform on source image and enhanced image to form new visible image and infrared image. Finally, the new visible image, infrared image and detail layers are superimposed to obtain the fusion result.The fused result not only contains the

contours information of the source image, but also retains the texture details in a relatively complete way. The proposed algorithm is superior to the existing algorithms in subjective and objective evaluation. Experimental results show that the image fused by the proposed algorithm is of high resolution and can better display the edge and

detailed texture information. The manuscript has been revised in the reviewers' opinion and is acceptable.

7. PLOS authors have the option to publish the peer review history of their article (what does this mean?). If published, this will include your full peer review and any attached files.

Reviewer #1: No

Reviewer #3: **Yes: **Dongming Zhou

---

## [Editor Report · Acceptance letter]

11 Jan 2021

PONE-D-20-32911R2 

Fusion algorithm of visible and infrared image based on anisotropic diffusion and image enhancement 

Dear Dr. Liu:

I'm pleased to inform you that your manuscript has been deemed suitable for publication in PLOS ONE. Congratulations! Your manuscript is now with our production department. 

Kind regards, 

on behalf of

Dr. Gulistan Raja 

Academic Editor

PLOS ONE